# Tsc2 disruption in mesenchymal progenitors results in tumors with vascular anomalies overexpressing *Lgals3*

Peter J Klover[1], Rajesh L Thangapazham[1], Jiro Kato[2], Ji-an Wang[1], Stasia A Anderson[2], Victoria Hoffmann[3], Wendy K Steagall[2], Shaowei Li[1], Elizabeth McCart[1], Neera Nathan[1,2], Joshua D Bernstock[1], Matthew D Wilkerson[4,5], Clifton L Dalgard[4,5], Joel Moss[2], Thomas N Darling[1]*

[1]Department of Dermatology, Uniformed Services University of the Health Sciences, Bethesda, United States; [2]Cardiovascular and Pulmonary Branch, National Heart, Lung, and Blood Institute, National Institutes of Health, Bethesda, United States; [3]Diagnostic and Research Services Branch, National Institutes of Health, Bethesda, United States; [4]Department of Anatomy Physiology and Genetics, Uniformed Services University of the Health Sciences, Bethesda, United States; [5]The American Genome Center, Uniformed Services University of the Health Sciences, Bethesda, United States

*For correspondence: thomas. darling@usuhs.edu

Competing interests: The authors declare that no competing interests exist.

**Abstract** Increased mTORC1 signaling from *TSC1/TSC2* inactivation is found in cancer and causes tuberous sclerosis complex (TSC). The role of mesenchymal-derived cells in TSC tumorigenesis was investigated through disruption of *Tsc2* in craniofacial and limb bud mesenchymal progenitors. Tsc2cKO[Prrx1-cre] mice had shortened lifespans and extensive hamartomas containing abnormal tortuous, dilated vessels prominent in the forelimbs. Abnormalities were blocked by the mTORC1 inhibitor sirolimus. A Tsc2/mTORC1 expression signature identified in Tsc2-deficient fibroblasts was also increased in bladder cancers with *TSC1/TSC2* mutations in the TCGA database. Signature component *Lgals3* encoding galectin-3 was increased in Tsc2-deficient cells and serum of Tsc2cKO[Prrx1]-cre mice. Galectin-3 was increased in TSC-related skin tumors, angiomyolipomas, and lymphangioleiomyomatosis with serum levels in patients with lymphangioleiomyomatosis correlating with impaired lung function and angiomyolipoma presence. Our results demonstrate Tsc2-deficient mesenchymal progenitors cause aberrant morphogenic signals, and identify an expression signature including *Lgals3* relevant for human disease of *TSC1/TSC2* inactivation and mTORC1 hyperactivity.

## Introduction

Mechanistic target of rapamycin complex 1 (mTORC1) is a central regulator of cell growth and metabolism (*Laplante and Sabatini, 2012*). Activation of mTORC1, caused by dysregulated upstream signaling through phosphoinositide 3-kinase (PI3K), PTEN, AKT, and TSC1-TSC2, is observed in cancers, hamartoma syndromes such as tuberous sclerosis complex (TSC), and vascular anomalies (*Dibble and Cantley, 2015*; *Krymskaya and Goncharova, 2009*; *Nathan et al., 2017*). The supposition that many of the pathological abnormalities in these conditions arise from increased signaling through mTORC1 is supported by response to treatment using mTOR inhibitors such as sirolimus and everolimus, particularly for tumors in TSC (*Bissler et al., 2008*; *Krueger et al., 2010*; *McCormack et al., 2011*; *Taveira-DaSilva et al., 2011*). However, clinical responses may be

**eLife digest** Tuberous sclerosis complex is a genetic condition that causes non-cancerous tumours with lots of blood vessels. It is caused by mutations that inactivate either of two genes known as *TSC1* and *TSC2*. A signalling molecule called mTOR also contributes to the disease, and drugs that block its activity provide some relief for patients. However, mTOR regulates a wide variety of molecules and so researchers are looking for which ones are responsible for the formation of the tumours.

Mesenchymal cells produce bone, muscle and other structural tissues in the body. They also support the formation of blood vessels. Mice – which are often used as model animals in health research – also have mesenchymal cells and a gene that is very similar to the human *TSC2* gene (known as *Tsc2*). Klover et al. hypothesized that disrupting the *Tsc2* gene specifically in the mesenchymal cells of mice may mimic aspects of tuberous sclerosis complex in humans.

The experiments show that disrupting *Tsc2* in mesenchymal cells does indeed mimic features of the human disease; the mice had shorter lifespans and they developed many tumours with dilated and winding blood vessels. Treating the mice with a drug that inhibits mTOR caused the tumours to shrink. Further experiments show that the loss of *Tsc2* alters the production of many proteins involved metabolism, cell growth and sensing the levels of oxygen. For example, mouse cells that lack *Tsc2* produce more of a protein called galectin-3, which appears to help blood vessels and tumours to grow in cancers.

Klover et al. also studied tumours from patients with tuberous sclerosis complex and a lung disease that is caused by mutations in *TSC2* (called lymphangioleiomyomatosis). The experiments found that many tumours produce higher levels of galactin-3 than normal cells. Bladder cancers with mutations in *TSC1* or *TSC2* also had higher levels of galactin-3, suggesting that other diseases linked with mutations in these genes may also result in increased production of galactin-3.

The findings of Klover et al. suggest that galectin-3 may be a useful marker to assess the severity of tuberous sclerosis complex, lymphangioleiomyomatosis and to detect cancers with mutations in *TSC1* or *TSC2*. The next step is to investigate whether galectin-3 alters blood vessels and tumour growth in these conditions.

inadequate and require lifelong treatment (*Taveira-DaSilva and Moss, 2015*). There is a need for greater understanding of potential downstream effectors of mTORC1 that may represent new targets for treatment and/or markers of disease severity.

TSC is a familial tumor syndrome characterized by highly vascular, hamartomatous tumors in multiple organs including the skin (eg. facial angiofibromas), kidneys (angiomyolipomas), and lungs (lymphangioleiomyomas). Facial angiofibromas can be disfiguring and their highly vascular nature makes them prone to bleeding with minimal trauma (*Darling et al., 2010*). Angiomyolipomas (AMLs) are a leading cause of death in patients with TSC due to hemorrhage and renal failure (*Franz et al., 2010*; *Byard et al., 2003*). They have large, tortuous, and thick-walled vessels that may lack elastin, making them prone to aneurysms and life-threatening hemorrhage (*Byard et al., 2003*; *Bissler and Kingswood, 2004*; *Tweeddale et al., 1955*). Lymphangioleiomyomatosis (LAM) involves a proliferation of abnormal smooth-muscle like cells that invade the axial lymphatics and lung to cause lymphangioleiomyomas and cystic lung disease, respectively (*Taveira-DaSilva and Moss, 2015*). LAM occurring in the absence of TSC, called sporadic LAM (S-LAM), is also associated with the development of AMLs (*Taveira-DaSilva and Moss, 2015*). The lymphangiogenic factor VEGF-D is elevated in LAM, correlates with disease severity and response to treatment, and is associated with lymphatic involvement (*McCormack et al., 2011*; *Glasgow et al., 2009*; *Young et al., 2013*; *Seyama et al., 2006*; *Budde et al., 2016*; *Malinowska et al., 2013*). VEGF-D is also increased in a mouse model of LAM (*Goncharova et al., 2012*).

TSC and S-LAM are caused by inactivating mutations in either *TSC1* or *TSC2* (*Cheadle et al., 2000*), genes that are also mutated in some cancers, particularly bladder carcinoma (*Sjödahl et al., 2011*; *Pymar et al., 2008*; *Guo et al., 2013*). Proteins encoded by the *TSC1* and *TSC2* genes, TSC1 (also known as hamartin) and TSC2 (aka tuberin), suppress mTORC1 signaling by forming a ternary

complex with TBC1D7 that suppresses RHEB-mediated activation of signaling through mTORC1 by converting RHEB-GTP to RHEB-GDP (*Dibble and Cantley, 2015*; *Dibble et al., 2012*). Loss of function of either TSC1 or TSC2 inhibits RHEB inactivation, leading to hyperactive mTORC1 signaling (*Dibble et al., 2012*; *Huang and Manning, 2008*). mTORC1 incorporates signals from growth factor signaling, especially through the PI3K-AKT pathway, and acts as a sensor of cellular stress, levels of amino acids, energy, and oxygen to mediate its downstream effects (*Dibble and Cantley, 2015*).

TSC1 or TSC2 loss of function and subsequent mTORC1 activation, which drives tumor formation and vascular changes, have been investigated using rodent models, exploiting a spontaneous mutation in *Tsc2* in the Eker rat, or using targeted disruption of *Tsc1* or *Tsc2* in mice. In the Eker rat, renal tumors develop with 100% penetrance and these rats additionally develop pituitary adenomas, uterine leiomyomas, and splenic tumors (*Eker, 1954*; *Yeung et al., 1994*). In mice, homozygous disruption of *Tsc1* or *Tsc2* is lethal during embryogenesis, and heterozygous $Tsc1^{+/-}$ and $Tsc2^{+/-}$ mice develop renal cystadenomas, liver hemangiomas, and infrequently, paw angiosarcomas (*Kobayashi et al., 1999*; *Onda et al., 1999*; *Kobayashi et al., 2001*; *Kwiatkowski et al., 2002*). Several models of Tsc1 deficiency have shown its role in the development of vascular abnormalities. Conditional disruption of *Tsc1* in vascular smooth muscle cells resulted in mice with vascular smooth muscle hyperplasia and hypertension (*Malhowski et al., 2011*; *Houssaini et al., 2016*). Deletion of Tsc1 expression specifically in endothelial cells using Tie2-cre led to embryonic lethality with embryos displaying a disorganized vascular network with edema and hemorrhage (*Ma et al., 2014*). By using an inducible Tie2-cre to disrupt *Tsc1* in postnatal mice, cutaneous lymphangiosarcomas and Prox1-positive thin-walled vascular channels developed with an increase in VEGFA levels within cutaneous tumors (*Sun et al., 2015*). Another model of *Tsc1* conditional disruption using Darpp32-cre developed kidney cysts by 8 weeks of age and angiosarcomas within the digits visible by postnatal day 21 (*Leech et al., 2015*). These models have demonstrated that Tsc1 deficiency in endothelial cells induces the formation of tumors by a mechanism involving mTORC1, but additional models are needed to replicate the pathological vascular changes observed in larger vessels in TSC, particularly since analysis of human AMLs has demonstrated TSC2 loss and mTORC1 activation in vessel walls (*Karbowniczek et al., 2003*).

Our previous research has demonstrated that TSC skin lesions usually contain *TSC2*-deficient fibroblast-like cells with hyperactive mTORC1 signaling (*Li et al., 2005, 2008*; *Tyburczy et al., 2014*). These fibroblast-like cells, upon incorporation into xenografts, resulted in skin with increased blood vessel size and number (*Li et al., 2011*). As we previously determined that dermal but not epidermal TSC2 loss occurs in human TSC skin samples (*Li et al., 2008*), we hypothesized that conditional disruption of mouse *Tsc2* in mesenchymal cells including dermal cells of the skin would be sufficient to induce highly vascular skin tumor formation and produce a source of Tsc2-deficient cells that could be used to discover factors that contribute to TSC tumorigenesis or have potential as diagnostic or prognostic markers of disease. Here we report the generation of a mouse model with a *Prrx1-cre* transgene (*Logan et al., 2002*) to disrupt a conditional *Tsc2* allele (*Hernandez et al., 2007*) in the lateral plate mesoderm, which contains cells that give rise to the limb bud and craniofacial mesenchyme. In addition to its activity in dermal fibroblasts within these regions, Prrx1-cre is regionally expressed within adipocytes, chondrocytes, and osteoblasts, but not blood cells or endothelial cells (*Logan et al., 2002*; *Greenbaum et al., 2013*; *Calo et al., 2010*). Transcriptomic analysis of Tsc2-deficient neonatal dermal fibroblasts from these mice in the presence or absence of sirolimus was used to screen for TSC2-dependent genes that could be indicators for TSC2 deficiency and TSC. This expression signature of TSC2 loss was increased in human bladder cancer with *TSC1* or *TSC2* mutation. Gal-3, a pro-angiogenic lectin, was increased in mouse and human samples with TSC2 deficiency and Gal-3 serum levels correlated with LAM severity and the presence of AMLs in a cohort of patients with LAM.

## Results

### Tsc2cKO[Prrx1-cre] mice efficiently disrupt *Tsc2* in mesenchymal cells and have reduced lifespan

We generated mice with disruption of *Tsc2* in mesenchymal progenitor cells by crossing mice with a conditional *Tsc2* allele (*Hernandez et al., 2007*) (*Tsc2^fl*) with mice carrying the *cre recombinase*

transgene driven by a *Prrx1* enhancer element (*Logan et al., 2002*) and with mice carrying an EYFP fluorescent reporter gene (*Srinivas et al., 2001*) to track $Tsc2^{-/-}$ (KO) cells. Mice containing homozygous $Tsc2^{fl}$ allele and heterozygous *Prrx1-cre* transgene (herein, Tsc2cKO$^{Prrx1-cre}$ mice) were born live and in expected ratios from crosses (see Materials and methods section). EYFP expression was detected in the limbs, ventral skin and craniofacial regions (*Figure 1A*).

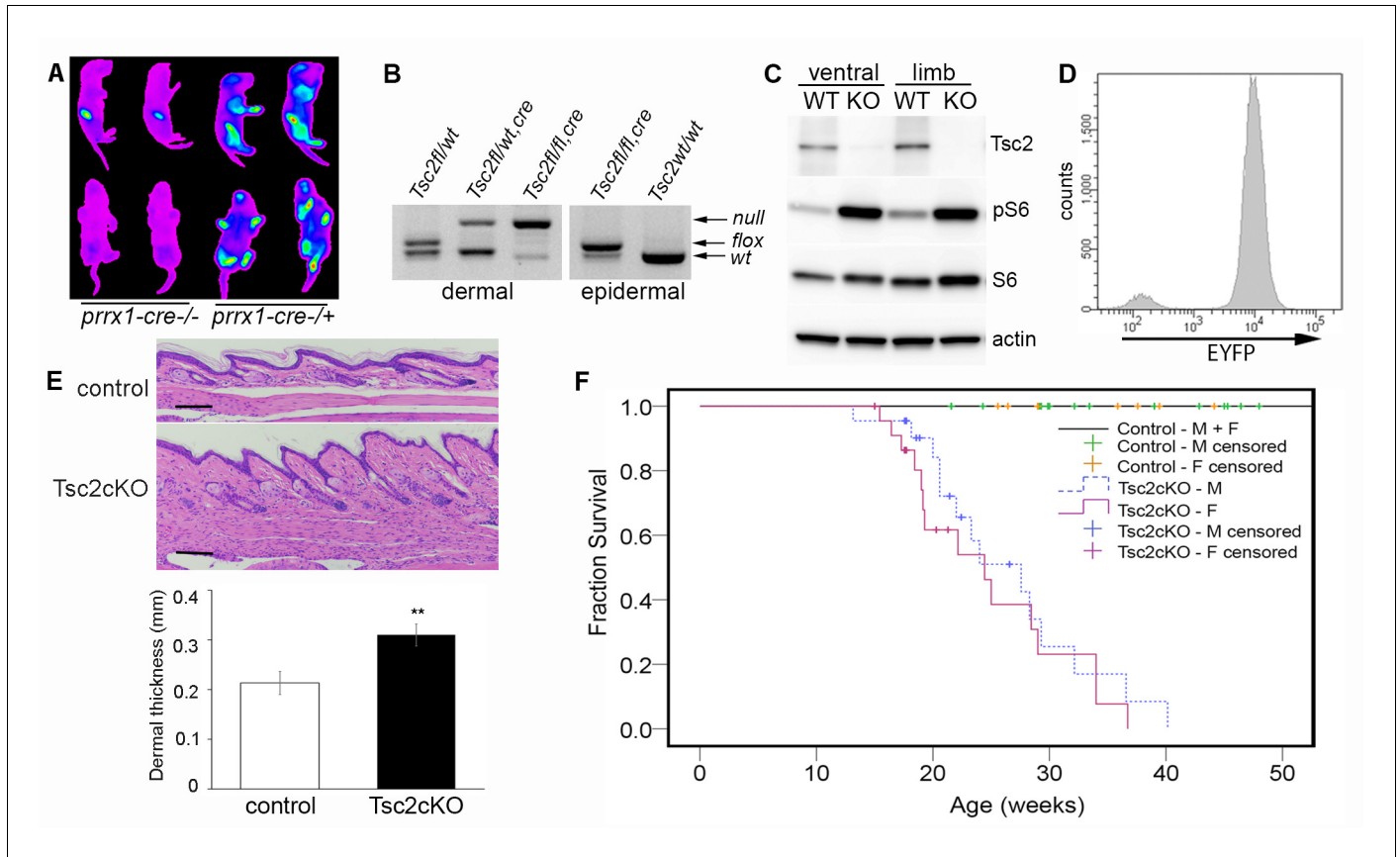

**Figure 1.** Characterization of Tsc2cKO$^{Prrx1-cre}$ mice. (**A**) In vivo imaging of EYFP fluorescence from neonates without (left 2 pups) or expressing *Prrx1-cre* (right 2 pups). Fluorescence was observed in stomach from milk. (**B**) Genotype of neonatal dermal fibroblasts from *Prrx1-cre* expressing mice (left panel) and neonatal epidermis (right panel). Labels represent genotype that was observed from tail DNA, which does not express *Prrx1-cre* or contain recombined (*Tsc2*) alleles. (**C**) Western blot of protein from neonatal dermal fibroblasts isolated from ventral skin or limbs of Tsc2cKO$^{Prrx1-cre}$ (KO) or $Tsc2^{fl/fl}$ controls (WT) probed with indicated antibodies. Similar results seen with greater than 10 cell lines. (**D**) Flow cytometry of cultured EYFP-expressing neonatal limb dermal fibroblasts. Approximately 95% of cells expressed EYFP. (**E**) Histology of skin of Tsc2cKO mice demonstrating greater dermal thickness and cellularity than controls. Scale bar, 0.1 mm. Dermal thickness of dorsal forepaw skin in control and Tsc2cKO mice ages 2–5 months (control n = 9, Tsc2cKO n = 7). Data presented as mean ± SD, **p<0.001. (**F**) Kaplan-Meier survival analysis was used to determine the median survival of Tsc2cKO mice. The Tsc2cKO differed from control (p<0.001, log-rank test). Among Tsc2cKO mice, median survival for males was 28 weeks of age and for females 24 weeks, however this difference was not significant (p=0.392, log-rank test). Controls n = 28 (all 28 were censored), M Tsc2cKO n = 22 (8 censored), F Tsc2cKO n = 24 (9 censored). The source data for this figure are in *Figure 1—source data 1*, *2* and *3*.

The following source data and figure supplement are available for figure 1:

**Source data 1** Source data for *Figure 1E*.
**Source data 2.** Source data for *Figure 1F*.
**Source data 3.** Source data for *Figure 1F*.
**Figure supplement 1.** Facial and skeletal phenotype of Tsc2cKO$^{Prrx1-cre}$ (cKO) mice.

Analysis by PCR showed that nearly all limb skin fibroblasts, but not epidermal cells, contained the recombined *Tsc2^fl* allele (herein, *Tsc2^-* allele) (*Figure 1B*), confirming mesenchymal specificity. Western blot analysis of fibroblasts from ventral and limb skin of Tsc2cKO^Prrx1-cre mice (KO) showed nearly undetectable Tsc2 protein in KO as well as increased phosphorylation of S6, indicating mTORC1 hyperactivation (*Figure 1C*). Flow cytometry analysis of cultured KO neonatal leg skin fibroblasts indicated that approximately 95% of cells expressed EYFP (*Figure 1D*), corresponding to the dramatically reduced level of Tsc2 protein observed in these cells. Postnatal Tsc2cKO^Prrx1-cre mice had shorter and thicker extremities, thicker bones of the cranium and limbs and a bulbous snout (*Figure 1—figure supplement 1*). Histological examination of dorsal forepaw (*Figure 1E* and *Figure 1—source data 1*) and whisker pad skin (*Figure 1—figure supplement 1b*) from adult Tsc2cKO mice revealed a thickened, hypercellular dermis. Survival was reduced in Tsc2cKO^Prrx1-cre mice with median survival of approximately 24 weeks, and no significant gender difference (*Figure 1F* and *Figure 1—source data 2*). Blood counts and serum chemistry were performed on 17 week old Tsc2cKO^Prrx1-cre mice to explain early death and revealed evidence of anemia, however serum markers of kidney and liver function were not different from controls (*Figure 1—source data 3*).

## Tsc2cKO^Prrx1-cre mice develop vascular hamartomas in multiple tissues

A growth on the volar surface of forepaws of all Tsc2cKO^Prrx1-cre mice was visible beginning at about 3 weeks of age (*Figure 2A*). Serial magnetic resonance imaging (MRI) analysis, done at 4, 8 and 12 weeks of age in Tsc2cKO^Prrx1-cre mice (n = 6) and controls (WT) of similar age, showed fluid-filled spaces and nodular masses in some kidneys as early as 4 weeks and in kidneys of 6 of 6 mice by 12 weeks (*Table 1*, *Figure 2B* green arrow). The spleen became enlarged with poorly defined internal structure/patterning in 3 of 6 mice by 8 weeks, and 5 of 6 mice by 12 weeks (*Table 1* and *Figure 2B*, blue arrow). A fluid-containing abnormality in the subcutaneous layer appeared in the shoulder region near the neck and axilla, which increased in size and frequency with age to become nearly universal by 12 weeks of age (*Table 1* and *Figure 2B*, pink arrows). Changes in the liver were less frequently detected by MRI, with dark speckling observed in 1/6 animals by 12 weeks of age.

Kidneys showed grossly visible cystic lesions (*Figure 2C*) that were microscopically apparent in 9 of 11 mice. EYFP expression, an indication of KO cells, was observed in cyst wall epithelial cells (*Figure 2—figure supplement 1B*). Livers developed vascular growths in 8 of 11 Tsc2cKO^Prrx1-cre mice (*Figure 2C*), and contained EYFP expression in vessel intima (*Figure 2—figure supplement 1D*). All adult Tsc2cKO^Prrx1-cre mice developed enlarged spleens with grossly visible tumor nodules (*Figure 2D*, left). Results of microscopic evaluation of 5 male and 6 female Tsc2cKO^Prrx1-cre mice at an average age of 24 weeks is presented in *Table 2*.

By approximately 12 weeks of age, swelling in the upper thorax near the neck/shoulder was grossly visible. Post-mortem examination revealed cysts involving the proximal forelimbs and upper chest (*Figure 2D* right). Angiography demonstrated large, tortuous vessels both in the anterior and posterior regions of Tsc2cKO^Prrx1-cre mice (*Figure 2—figure supplement 2*). In addition the cystic masses also appeared to have a lymphatic component based on fluid collected from subcutaneous axillary masses outside the thorax. This fluid was pink and cloudy (*Figure 2—figure supplement 3*) and contained higher triglyceride than paired serum samples from these mice, suggesting a contribution of chylous fluid (*Figure 2—figure supplement 3*).

Histological examination of upper forelimbs showed vascular anomalies with tortuous, dilated blood and lymphatic vessels amid abnormal adipose tissue and skeletal muscle (*Figure 2E*). Different size proliferations of blood vessels was observed including some containing hyalinized tunica media, smooth muscle hyperplasia, and smooth muscle surrounded by proliferations of smaller capillary-like vessels (*Figure 2Ei*). Large, dilated vessels, many with thick walls, contained thin, fragmented elastic fibers reminiscent of those observed in human AML (*Figure 2Eii*, *Figure 2—figure supplement 4*). Adipose tissue was reduced in the subcutaneous space of Tsc2cKO^Prrx1-cre mice. In contrast, hamartomas found in shoulder frequently contained abundant adipose tissue containing plump nuclei, increased cytoplasm, and either single or multiple fat vesicles (*Figure 2Eiii*). Abnormal lymphatics contained tortuous lymph node sinuses and vessels lined with plump endothelial cells and surrounded by increased collagen (*Figure 2Eiv*). Abnormal lymphatic vessels expressed typical lymphatic markers (VEGFR3, LYVE1), and also expressed EYFP indicating loss of Tsc2 expression (*Figure 2—figure supplement 5*).

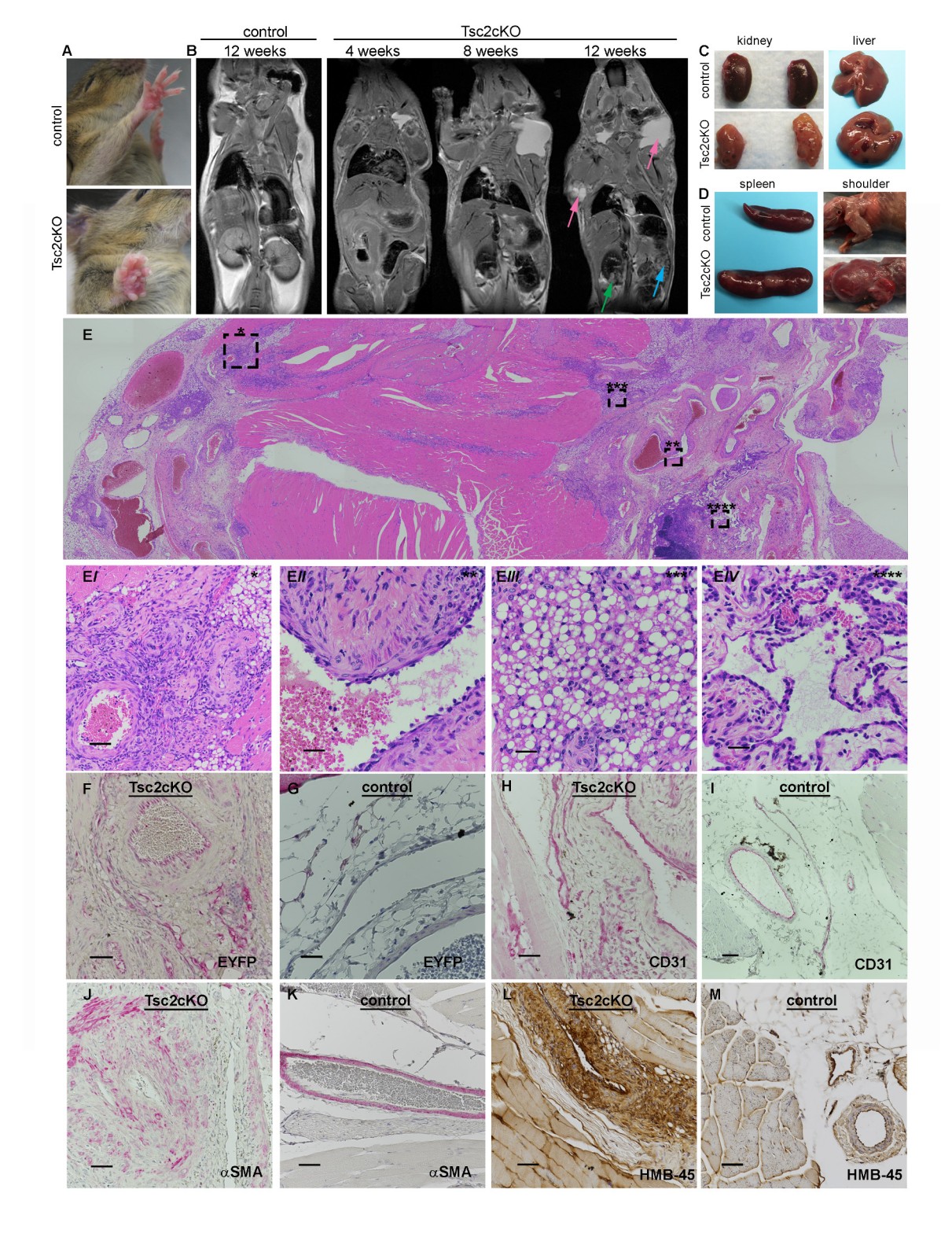

**Figure 2.** Gross and histopathology of Tsc2cKO[Prrx1-cre] mice (Tsc2cKO). (**A**) Forepaw growth in adult Tsc2cKO. (**B**) Full-body MRI 2D images of 12 week control and serial-imaged Tsc2cKO mice. High signal intensity was detected in upper chest/shoulder (pink arrows). Other abnormalities are seen in kidney (small cysts, green arrow) and spleen (enlargement and irregular patterning, blue arrow). (**C**) Gross appearance of kidney and liver from 4 to 5 month old control (upper panels) and adult Tsc2cKO mice (lower panels) showing typical tumors. (**D**) Gross appearance of spleen and shoulder region

*Figure 2 continued on next page*

*Figure 2 continued*

from 4 to 5 month old control (upper panels) and adult Tsc2cKO mice (lower panels). (E) Histologic appearance of shoulder and axillary region of Tsc2cKO mouse. (E*i*) Hamartomatous region with abnormal vessels and abundant fat. (E*ii*) Abnormally large blood vessel with thick, fibrotic vessel wall and endothelial dysplasia. (E*iii*) Lipomatous area contains adipose-like cells with variably-increased eosinophilic cytoplasm. (E*iv*) Lymphatic dysplasia near lymph node. Immunohistochemical studies including anti-GFP (F, G) anti-CD31 (H, I), anti-alpha-smooth muscle actin (SMA) (J, K) and anti-HMB-45 (L, M) of forelimb hamartomas in Tsc2cKO mice (F, H, J, L) and forelimb vessels of control mice (G, I, K, M). IHC staining of controls and Tsc2cKO tissue was consistent in sections from at least n = 4 mice. Scale bars: 0.5 mm for E; 0.05 mm for E*i*, 0.025 mm for E*ii-*E*iv*, G and K; 0.05 mm for F, H, I J, L and M.

The following figure supplements are available for figure 2:

**Figure supplement 1.** IHC of kidney and liver tumors.

**Figure supplement 2.** MRI angiography of Tsc2cKO[Prrx1-cre] mice.

**Figure supplement 3.** Triglyceride and cholesterol analysis of fluid from axillary mass outside thorax.

**Figure supplement 4.** Elastin staining of human and mouse blood vessels.

**Figure supplement 5.** Upper forelimb lymphatic hamartoma in Tsc2cKO[Prrx1-cre] mice.

**Figure supplement 6.** Histology of Tsc2cKO forepaw and spleen tumors.

To detect KO cells in sections of forelimbs, EYFP expression was examined using anti-GFP antibodies. EYFP expression was found throughout the hamartomatous tumors of upper forelimbs and frequently seen in vessels including endothelial cells (*Figure 2F*). In contrast, control vessels from the upper forelimbs showed EYFP expression in connective tissue fibroblasts and mural cells of vessels, but EYFP staining was not observed in luminal cells (*Figure 2G*). Within thickened vessels, staining using the blood vessel endothelial marker CD31 highlighted abnormal plump endothelial cells and slit-like spaces in the vessel walls (*Figure 2H*), features not evident in control vessels (*Figure 2I*). These abnormal vessel walls contained greater numbers of smooth muscle cells in disorganized arrangement (*Figure 2J*) than control vessels from the upper forelimb (*Figure 2K*). Hamartomas also expressed HMB-45, an immunohistochemical marker of LAM and AMLs, in the anomalous vessels of upper forelimbs (*Figure 2L*) but not control vessels (*Figure 2M*).

Tumors of the forepaws and spleen showed abnormal proliferations of blood vessels without accompanying alterations of fat and lymphatics as in the shoulder region. Vascular anomalies of the volar forepaw spanned the bone to the dermis (*Figure 2—figure supplement 6A*) with numerous abnormally thick, hyalinized vessels with smooth muscle hyperplasia and many smaller vessels (*Figure 6—figure supplement 6Ai*). Grossly visible tumors were not observed in hind paws, possibly a reflection of the later Prrx1-cre expression observed in the developing hindlimbs compared to forelimbs (*Logan et al., 2002*). As in the shoulder tumors, EYFP expression was observed throughout the tumor including perivascular and vessel intima (*Figure 2—figure supplement 6B*). In contrast, similar age EYFP-expressing control mice (*Tsc2[fl/+], Prrx1-cre[+/-]*) displayed EYFP expression in forepaw fibroblasts and vessel mural cells, but not endothelium (*Figure 2—figure supplement 6C*).

**Table 1.** Numbers of mice with abnormalities in organs or upper chest (UC) based on serial MRI analysis of 6 Tsc2cKO mice at 4, 8, and 12 weeks of age to determine the onset of detectable hamartoma formation.

| Age | Kidney | Spleen | Liver | UC |
|---|---|---|---|---|
| 4 weeks | 1/6 | 0/6 | 0/6 | 3/6 |
| 8 weeks | 5/6 | 3/6 | 0/6 | 3/6 |
| 12 weeks | 6/6 | 5/6 | 1/6 | 5/6 |

**Table 2.** Gross and histopathological analysis of tumors present in 11 Tsc2cKO mice. Tumors were considered present by microscopic observation. The age range from mice in this group was 18–40 weeks with an average of 24 weeks. NL=normal, TA= tubular adenoma, VH= vascular hamartoma.

| Mouse | Kidney | Liver | Spleen | Paw |
|---|---|---|---|---|
| 1 | NL | VH | VH | NL |
| 2 | TA | VH | VH | VH |
| 3 | TA | VH | VH | VH |
| 4 | TA | VH | VH | VH |
| 5 | TA | VH | VH | VH |
| 6 | NL | NL | VH | VH |
| 7 | TA | VH | VH | VH |
| 8 | TA | VH | VH | VH |
| 9 | TA | NL | VH | VH |
| 10 | TA | NL | VH | VH |
| 11 | TA | VH | VH | VH |
| totals | 9/11 | 8/11 | 11/11 | 10/11 |

Smooth muscle hyperplasia was confirmed by staining for SMA (*Figure 2—figure supplement 6D*) and endothelial cells in both larger vessels and proliferations of smaller slit-like vessels stained positively with CD31 (*Figure 2—figure supplement 6E*).

Splenic tumor morphology appeared as nodules forming from large dysplastic blood vessels surrounded by areas containing extensive proliferations of smaller vessels and fibrosis (*Figure 2—figure supplement 6F, Fi*). EYFP expression was present in endothelial and perivascular cells, both of which were highly abundant within these tumors (*Figure 2—figure supplement 6G*). Similar to the forepaw, endothelial cells from EYFP-expressing controls did not express EYFP (*Figure 2—figure supplement 6H*). The large abnormal vessel walls were thickened with SMA-positive mural cells (*Figure 2—figure supplement 6I*) and surrounded by extensive proliferations of thin-walled smaller vessels with CD31-positive endothelial cells (*Figure 2—figure supplement 6J*).

## Tumors contain KO mesenchymal and endothelial cells and are sensitive to sirolimus treatment

To detect the presence of Tsc2-deficient cells, DNA from neonatal Tsc2cKO$^{Prrx1-cre}$ organs were tested using semi-quantitative 3-primer PCR. Compared to the expected recombination in the limbs, low levels of *Tsc2*$^-$ allele were present in kidney, liver and spleen of Tsc2cKO$^{Prrx1-cre}$ neonates, indicating the presence of KO cells before tumor formation (*Figure 3A* upper panel). Tumors of adult mice showed increased *Tsc2*$^-$ allele, providing evidence of an expansion of KO cells in tumor-containing tissues (*Figure 3A*, lower panel).

To confirm that EYFP organ-staining represented the presence of KO cells, real-time PCR copy number assays using primers within the conditional region of the *Tsc2*$^{fl}$ allele (exon 3) were used on isolated splenocytes from EYFP-fluorescing cells enriched by FACS (*Figure 3B* and *Figure 3—source data 1*). EYFP-positive cells had approximately 10% of the *Tsc2* exon 3 copy number as cells from WT spleen. Both *Tsc2* exon 6 (outside of the conditional region) and *Tsc1* copy number were unchanged in EYFP-positive cells. Therefore, EYFP positivity in this model is an accurate representation of KO cells. To enrich for mesenchymal cells, isolated splenocytes were cultured on plastic, using fibroblast growth medium. After enzymatic dissociation, cultured splenocytes were stained with fluorescent antibodies against the cell surface marker Thy1.2 (CD90.2), a known antigen of mesenchymal lineages. By flow cytometry, we found that about half of cultured splenocytes stain for CD90.2, including most EYFP expressing (KO) cells (*Figure 3C*). Consistent with IHC results, cultured cells from dissociated paw tumors contained an endothelial fraction that expressed EYFP (*Figure 3D*, middle right).

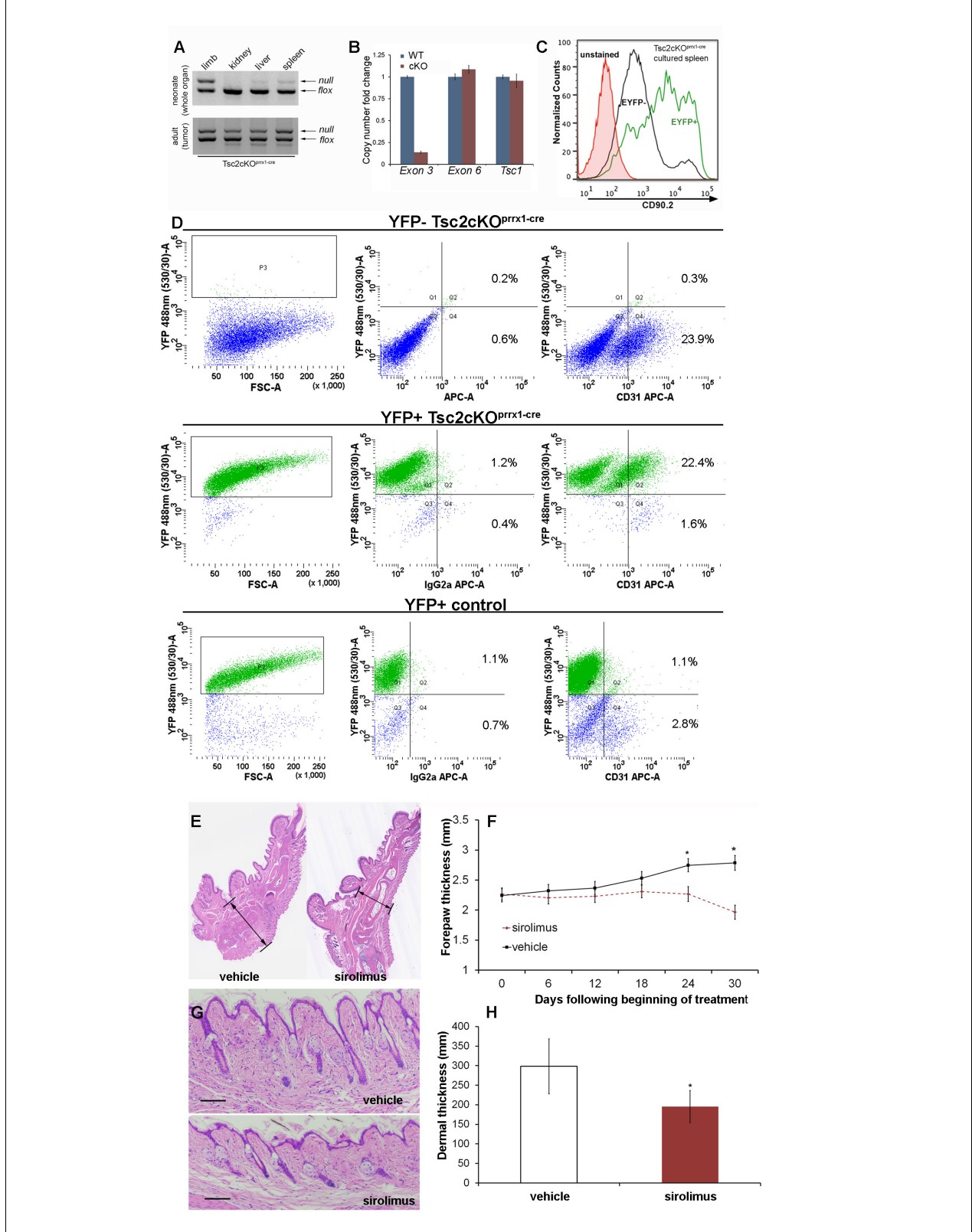

**Figure 3.** Hamartomas of Tsc2cKO[Prrx1-cre] (Tsc2cKO) mice contain both mesenchymal and endothelial KO cells and are sirolimus-sensitive. (**A**) 3-primer PCR detection of *Tsc2⁻*, *Tsc2fl* and *Tsc2⁺* alleles in neonatal organs (upper panel) and adult tumor tissue using genotyping primers (lower panel). Similar results found in three separate neonatal and adult Tsc2cKO mice. (**B**) Copy numbers of *Tsc2* exon 3 were reduced in DNA isolated from EYFP-expressing splenocytes enriched by FACS compared to DNA isolated from WT cells. *Tsc2* exon 6 and *Tsc1* copy number assays run as controls (n = 3

*Figure 3 continued on next page*

*Figure 3 continued*

from YFP+ spleen tumors). (C) The presence CD90.2 expression in EYFP-positive cultured splenocytes from cKO mice confirms Tsc2-deficient mesenchymal component containing Tsc2-deficient cells in spleen tumor. Similar results observed in two other cKO spleens. (D) Flow cytometry of dissociated and cultured forepaw tumor confirms that these tumors contain Tsc2-deficient CD31-positive cells. Upper: YFP-negative Tsc2cKO forepaw tumor cells. Middle: YFP-positive Tsc2cKO forepaw tumor cells. Lower: Dissociated and cultured EYFP-expressing cells from control forepaw. E, F: Reduction of tumor size by sirolimus. Sirolimus (5 mg/kg) or vehicle was injected IP every other day in Tsc2cKO mice starting at postnatal day 25 for 30 days. Forepaw thickness was then measured weekly by calipers and mice were sacrificed at day 30. (E) Histological sections of forepaws from 30 day treatment with vehicle (left) or sirolimus (right). (F) Forepaw thickness in cKO measured during treatment (day 24 *p=0.03, day 30 *p<0.001). G, H: Partial normalization of dermal thickness by sirolimus. (G) Histological image of dermis following 30 day sirolimus. Scale bar, 0.1 mm. (H) Measurement of dermal thickness following 30 day sirolimus (vehicle n = 8, sirolimus n = 10 mice, *p=0.003). Error bars for F and H indicate ± S.D. The source data for this figure are in *Figure 3—source datas 1–3*.

The following source data is available for figure 3:

**Source data 1.** Source data for *Figure 3B*.
**Source data 2.** Source data for *Figure 3H*.
**Source data 3.** Source data for *Figure 3F*.

These results demonstrate that anomalous vessels of the spleen and paw contain a combination of KO mesenchymal and endothelial cell components. However, in cells isolated from control EYFP-expressing forepaws, we observed very few EYFP-positive endothelial cells (*Figure 3D*, lower right). This is also consistent with published results of Prrx1-cre expression absent in bone marrow endothelial cells (*Greenbaum et al., 2013*).

To test if the thickened dermis and tumors in Tsc2cKO[Prrx1-cre] mice responded to mTORC1 inhibition, sirolimus treatment was started in recently weaned 25 day-old mice, for 30 days with alternate-day IP injection (5 mg/kg). Mice were sacrificed at approximately 7.5 weeks of age, two days following the last injection. Forepaw thickness was measured with calipers throughout the treatment course. In both male and female Tsc2cKO[Prrx1-cre] mice, a significant decrease in thickness across the middle of the forepaw was observed as compared to forepaws from vehicle-treated mice (*Figure 3E and F* and *Figure 3—source data 2*). Dermal thickness of the forepaws was reduced 35% compared to vehicle-treated controls (*Figure 3G and H* and *Figure 3—source data 3*). Post-treatment analysis revealed forepaw vascular hamartomas in 8 of 8 vehicle-treated Tsc2cKO[Prrx1-cre] mice compared to 1 of 10 sirolimus-treated mice. Kidney tumors and spleen tumors also responded to sirolimus treatment (*Table 3*).

## Transcriptome analysis of WT and Tsc2-deficient dermal fibroblasts with or without sirolimus

To identify novel Tsc2-dependent factors abnormally expressed in Tsc2-deficient dermal fibroblasts, RNA sequencing and gene expression data analysis were performed on the transcriptome of $Tsc2^{-/-}$ (KO, n = 3) and $Tsc2^{fl/fl}$ non-cre expressing control neonatal mouse leg skin fibroblasts (WT, n = 3). In untreated cells, 1387 genes were overexpressed in KO compared to WT, while 437 were overexpressed in WT compared to KO (false discovery rate <10%). These results are summarized as a

**Table 3.** Tsc2cKO mice were treated with either vehicle (n = 8) or 5 mg/kg sirolimus (n = 10) every other day for 30 days. Kidney, spleen, and paw were collected and analyzed by histological examination.

| Location | Vehicle | Sirolimus |
|---|---|---|
| Kidney | 4/8 | 0/10 |
| Spleen | 7/8 | 0/10 |
| Paw | 8/8 | 1/10 |

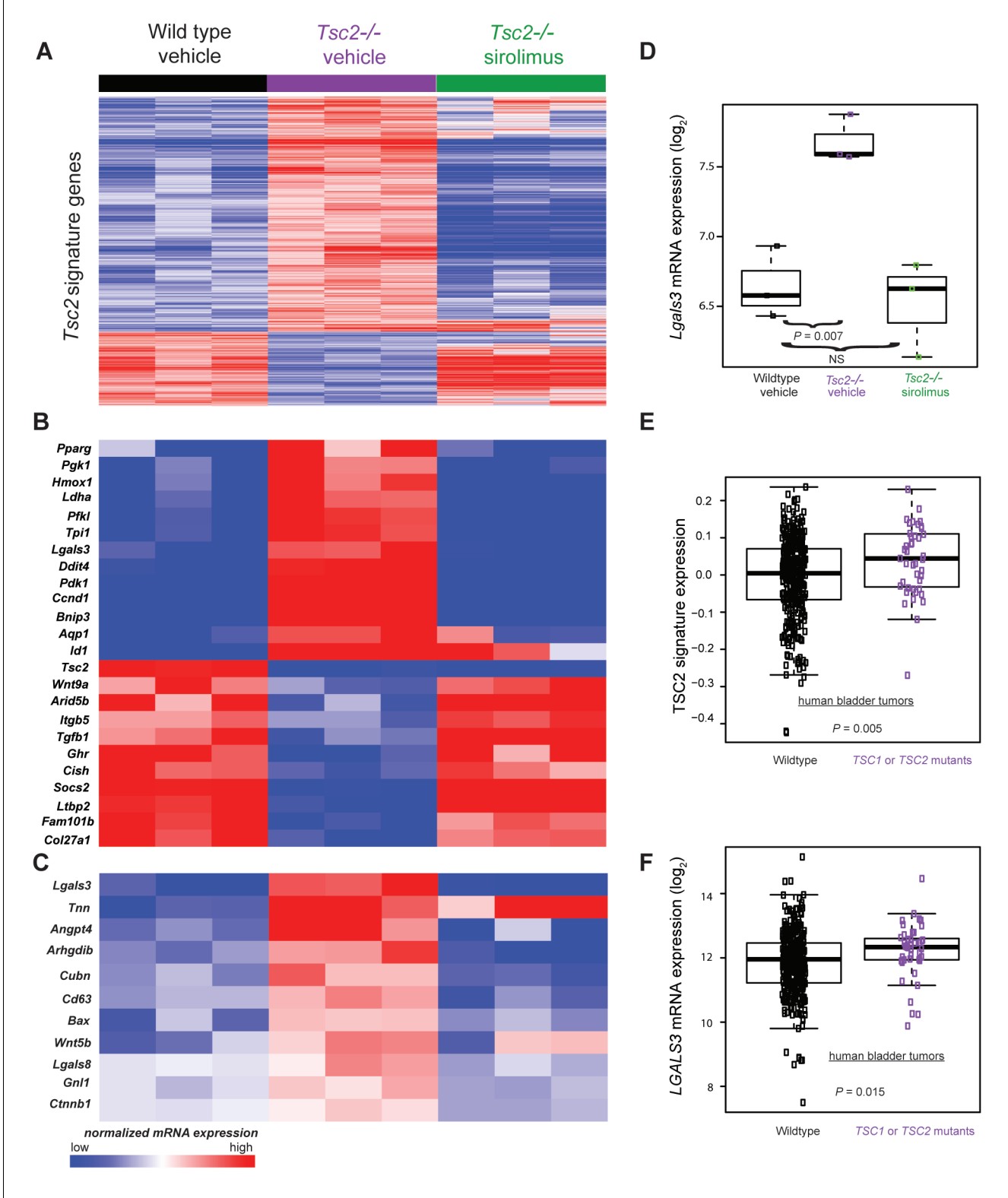

**Figure 4.** Transcriptomic analysis of mouse neonatal dermal fibroblasts identified Tsc2-dependent and mTORC1-dependent signature genes including *LGALS3*, whose mRNA is elevated in cancers with *TSC1* or *TSC2* mutations. (**A**) Heatmap of differentially regulated genes from n = 3 (each sample represents one neonate from a different litter of pups) WT, $Tsc2^{-/-}$ + vehicle, and $Tsc2^{-/-}$ + sirolimus-treated mouse neonatal dermal fibroblasts with FDR of <10%. Genes are centered to the median of wild type vehicle and $Tsc2^{-/-}$ *vehicle*. (**B**) Heatmap of selected genes from both Tsc2-dependent and mTORC1-dependent signature genes based on statistically over-represented gene ontology categories (p<0.001) including response to hypoxia,

*Figure 4 continued on next page*

Figure 4 continued

regulation of cell death, regulation of cell cycle, and glycolytic processes. (C) Heatmap of 11 genes overexpressed in $Tsc2^{-/-}$ and decreased by sirolimus treatment that matched to GO categories 'extracellular region' and 'signaling'. (D) Lgals3 expression in WT, $Tsc2^{-/-}$ + vehicle, and $Tsc2^{-/-}$ + sirolimus-treated dermal fibroblasts. (E) Mouse Tsc2-dependent gene expression signature is increased in human bladder cancers with non-silent mutations in either TSC1 or TSC2 (n = 43) compared with tumors containing WT TSC1 or TSC2 genes (n = 348, p=0.005). (F) LGALS3 mRNA expression is elevated in human bladder cancers with non-silent mutations in TSC1 or TSC2, p=0.015. Boxplot horizontal lines mark 25th, 50th, and 75th percentiles, whiskers extend to the furthest point less than or equal to 1.5 times the interquartile range. The source data for this figure are in *Figure 4— source data 1*, *2*, *3*, *4* and *5*.

The following source data is available for figure 4:

**Source data 1.** Source data for *Figure 4A*.
**Source data 2.** Source data for *Figure 4A*.
**Source data 3.** Source data-*Figure 4A*.
**Source data 4.** Source data-*Figure 4A*.
**Source data 5.** Source data 5-*Figure 4E and F*.

heatmap in *Figure 4A* (first 3 lanes vs middle 3 lanes, and see supplementary *Figure 4—source data 1* for the full list), which demonstrates high reproducibility of these differentially expressed genes among the dermal fibroblasts lines. Gene ontology enrichment analysis (see Materials and methods section) revealed that the signature in KO included genes involved in glucose metabolism, the regulation of cell cycle, and HIF1α responses.

To confirm that elevated mTORC1 signaling had an expected role in Tsc2-deficient KO fibroblasts, KO and WT neonatal dermal fibroblasts were treated with 20 nM sirolimus for 24 hr. Sirolimus treatment resulted in 7282 underexpressed genes in KO fibroblasts and 2567 overexpressed genes in KO fibroblasts (*Figure 4A* middle 3 lanes vs last 3 lanes, and *Figure 4—source data 2*). The effect of sirolimus was less in WT cells with 2852 underexpressed genes and 223 overexpressed mRNAs (*Figure 4—source data 3*). Genes effected by sirolimus in WT fibroblasts were nearly a subset of the genes effected by sirolimus in KO fibroblasts (overexpressed genes 87% in common; underexpressed genes 88% in common). Sirolimus corrected the effect of Tsc2-deficiency (WT vs. KO) for many mRNAs. Ninety-two percent (1275 of 1387) of genes overexpressed in KO compared to WT were also underexpressed after sirolimus treatment in KO. Likewise, 80% (349 of 437) of genes that were underexpressed in KO versus WT were also overexpressed after sirolimus treatment in KO (*Figure 4—source data 4*). These results indicate sirolimus had the expected effect of reversing much of the dysregulation caused by Tsc2 deficiency in these cells.

To identify individual genes that may be of relevance to the diagnosis and/or treatment of TSC, we screened sirolimus-sensitive genes also overexpressed in KO cells and manually reviewed the sirolimus-sensitive genes known to mediate developmental programs and/or angiogenesis as potential mediators of TSC pathogenesis (*Figure 4B*). Additionally, using PANTHER analysis (*Mi et al., 2017*, *2013*), we screened for genes that were present in both of the ontology categories 'extracellular region' and 'signaling', producing a list of 11 genes. (*Figure 4C*). The only transcript that matched both lists in *Figure 4B and C* was Lgals3, which codes for galectin-3 (Gal-3), a lectin with specificity for beta-galactoside moieties on glycoproteins and has been reported to play roles in angiogenesis and fibrosis (*Li et al., 2014*). Lgals3 was greater in KO than WT fibroblasts (p=0.007) and corrected by sirolimus (*Figure 4D*).

As TSC1 or TSC2 mutations occur in cancers, we sought to determine if the TSC2 loss-of-function expression signature was present in human bladder tumors since this tumor type often sustains inactivating TSC1 or TSC2 mutations. Utilizing published gene expression data of bladder tumors from The Cancer Genome Atlas (TCGA) (*Cancer Genome Atlas Research Network, 2014*), we calculated a signature score for 391 bladder tumors. Of these, 43 contained non-silent mutations in either TSC1 or TSC2 which included missense, nonsense, frame shift, splice site, in frame deletions, or 5'UTR mutations (*Figure 4—source data 5*). Signature scores were greater in tumors having non-

silent *TSC1* or *TSC2* mutations versus other bladder tumors (*Figure 4E*) indicating that these cancers have an identifiable gene expression signature derived from inactivation of *TSC1* or *TSC2*.

Interrogation of the TCGA cohort revealed that *TSC1* and *TSC2* mutant bladder tumors overexpressed *LGALS3* versus other bladder tumors. (*Figure 4F*), suggesting that Gal-3 may be an individual marker for bladder cancers containing inactivating mutations in *TSC1* or *TSC2*.

## Gal-3 is overproduced by mouse KO fibroblasts and is highly expressed in vascular hamartomas of Tsc2cKO^Prrx1-cre mice

In the forelimbs of WT mice, cells immunoreactive for Gal-3 were identified in larger vessels with staining seen in some skeletal muscle nuclei (*Figure 5A*, left), whereas in the forelimb tumor of Tsc2cKO^Prrx1-cre mice, areas of dense Gal-3 positivity were observed in both vascular and perivascular cells (*Figure 5A*, right). In the skin, Gal-3-positive cells were observed in both WT and Tsc2cKO^Prrx1-cre mice in the epidermis (*Figure 5B*). Additionally, Tsc2cKO^Prrx1-cre mice contained increased numbers of positively stained dermal fibroblasts (*Figure 5B*, lower). In early passage KO fibroblasts, both intracellular Gal-3 levels (*Figure 5C*) and secreted Gal-3 (*Figure 5D* and *Figure 5— source data 1*) were sharply increased, and 48 hr sirolimus [20 nM] treatment resulted in their partial normalization. Serum of adult Tsc2cKO^Prrx1-cre mice had 67% higher Gal-3 levels (p=0.015) than similar-age controls (*Figure 5E* and *Figure 5—source data 2*). In sirolimus-treated Tsc2cKO^Prrx1-cre mice, Gal-3 serum levels were decreased by 25% (p=0.036) (*Figure 5F* and *Figure 5—source data 2*).

## Gal-3 is overproduced in human TSC skin tumors and negatively correlates with lung function in patients with lymphangioleiomyomatosis (LAM)

Gal-3 immunostaining of normal-appearing control skin obtained from TSC patients showed positivity in the epidermis but very little in the dermis (*Figure 6A*). In contrast, TSC skin tumors had abundant Gal-3 positive dermal fibroblasts (*Figure 6B*). Gal-3 ELISA of supernatants from fibroblasts grown from TSC skin tumors released more Gal-3 than fibroblasts grown from normal-appearing skin (*Figure 6C* and *Figure 6—source data 1*). Western blot analysis of samples from four patients demonstrated higher intracellular protein levels of Gal-3 in TSC skin tumor fibroblasts than paired normal-appearing skin fibroblasts, although absolute levels of Gal-3 varied among patients (*Figure 6D*). Gal-3 staining in tissue sections from lungs of patients with LAM showed Gal-3 expression in LAM nodules (LAM, *Figure 6E*, *Figure 6—figure supplement 1*). In renal angiomyolipomas (AML), Gal-3 expression was observed in smooth muscle and adipose cells (*Figure 6F*, *Figure 6— figure supplement 2*). In patients with LAM not taking mTOR inhibitors, percent predicted one second forced expiratory volume (%FEV1) negatively correlated with Gal-3 levels, (*Figure 6G* and *Figure 6—source data 2* and 3). In serum from patients that were being treated with mTOR inhibitors, no correlation of %FEV1 was found with Gal-3 serum levels (*Figure 6—figure supplement 3*). Gal-3 levels were also analyzed with two-way factorial ANOVA in treatment naïve patients with or without a confirmed diagnosis of AML, grouped according to severe LAM (%FEV1 <80) and mild LAM (% FEV1 >80). Gal-3 levels were higher in the mild LAM group with AML compared to no AML suggesting AMLs were an additional source of serum Gal-3 (*Figure 6H-* and *Figure 6—source data 2* and 3). As it is known that BMI affects Gal-3 levels (*Weigert et al., 2010*), we tested if our results could be explained by differences in BMI of the patients whose samples we tested. There was a significant positive correlation between BMI and galectin-3 in patients without mTOR inhibitor (r = 0.315, p=0.013). After adjusting for BMI, the partial correlation coefficient between FEV1 and galectin-3 in patients without mTOR inhibitor was still significant (partial correlation r = −0.366, p=0.004). No significant difference in Gal-3 levels was observed between patients with LAM not taking mTOR inhibitor (4649 ± 1980 pg/mL, n = 64) and normal subjects (5023 ± 1646 pg/mL, n = 25).

## Discussion

Disruption of *Tsc2* in mesenchymal progenitors caused extensive and remarkable vascular abnormalities, including dilated, thickened, and tortuous blood vessels in the limbs and neck along with dilated lymphatic vessels and large lymphatic cysts in the axillary and neck regions. In the forepaws, hamartomatous tumors with vascular anomalies formed that were comparable to angiosarcomas or lymphangiosarcomas reported in *Tsc1* or *Tsc2* genetic mouse models (*Onda et al., 1999*;

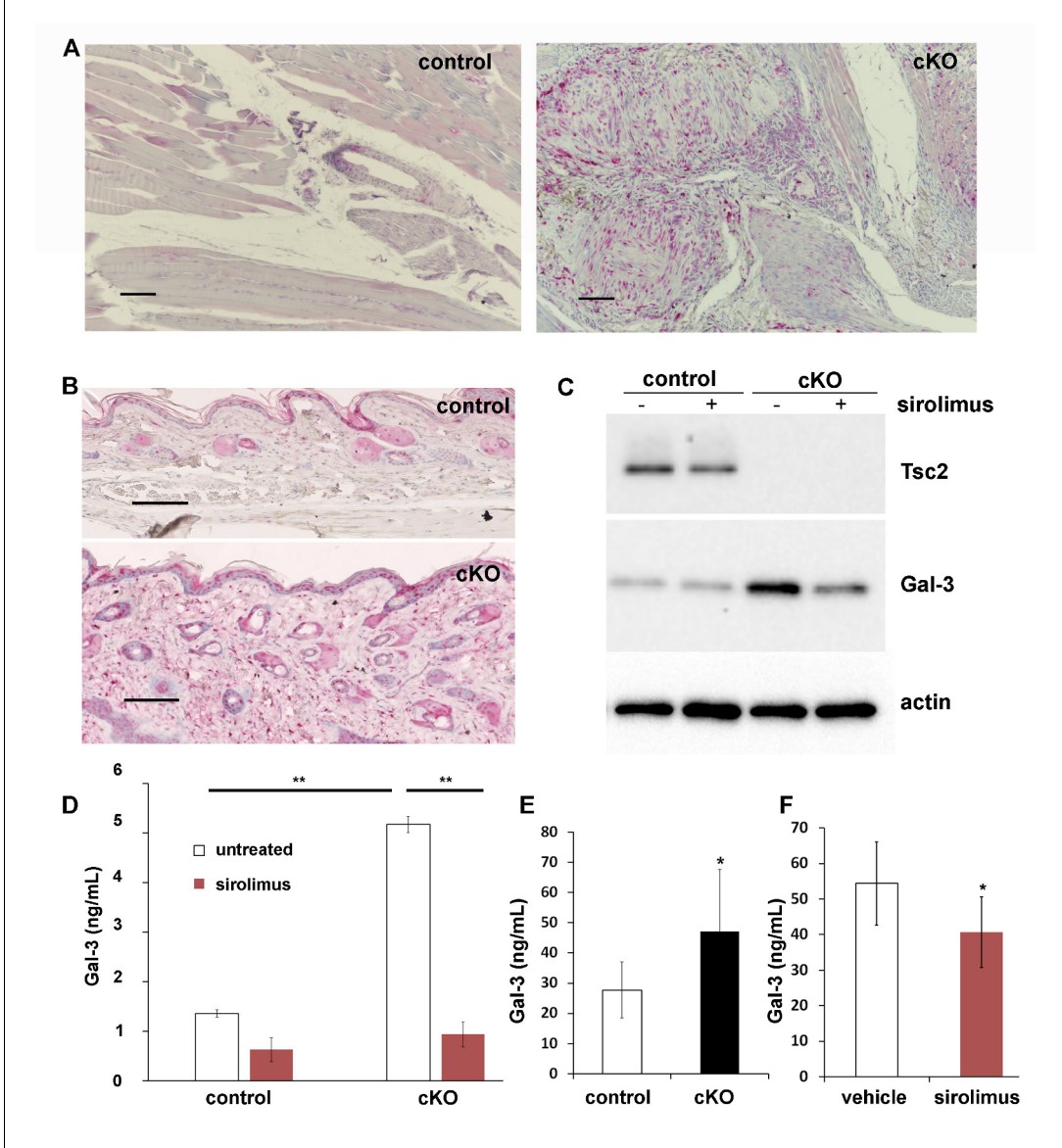

**Figure 5.** Increased production and secretion of Gal-3 in Tsc2cKO[Prrx1-cre] mice (cKO), which is partially under the control of mTORC1. (**A**) Gal-3 immunostaining of forelimb tissues of control and cKO. Hamartoma of cKO mice shows Gal-3 positive staining within vascular and perivascular cells. (**B**) Gal-3 immunostaining of forepaw dermis from control and cKO mice. (**C**) Western blot of neonatal WT and KO dermal fibroblasts untreated or treated with 20 nM sirolimus for 48 hr. Blots were probed with antibodies to Tsc2, Gal-3 and actin. (**D**) Gal-3 secretion from 48 hr culture supernatants of WT and KO neonatal dermal fibroblasts treated as indicated. Gal-3 levels were measured by a mouse ELISA assay (n = 3 cell lines for each group). *p<0.05. (**E**) A significant increase (p=0.004) in serum levels of Gal-3 from adult Tsc2cKO[Prrx1-cre] mice (n = 11) compared to control mice (n = 15) was observed. (**F**) Serum from 8 week-old, 30 day sirolimus-treated treated mice (n = 8 mice) showed a significant (p=0.04) decrease in Gal-3 compared to controls (n = 6 mice). Error bars for D-F indicate ± S.D. Scale bars in A and B are 0.1 mm. The source data for this figure are in *Figure 5—source data 1*, *2*.

The following source data is available for figure 5:

**Source data 1.** Source data-*Figure 5D*.
**Source data 2.** Source data-*Figure 5E and F*.

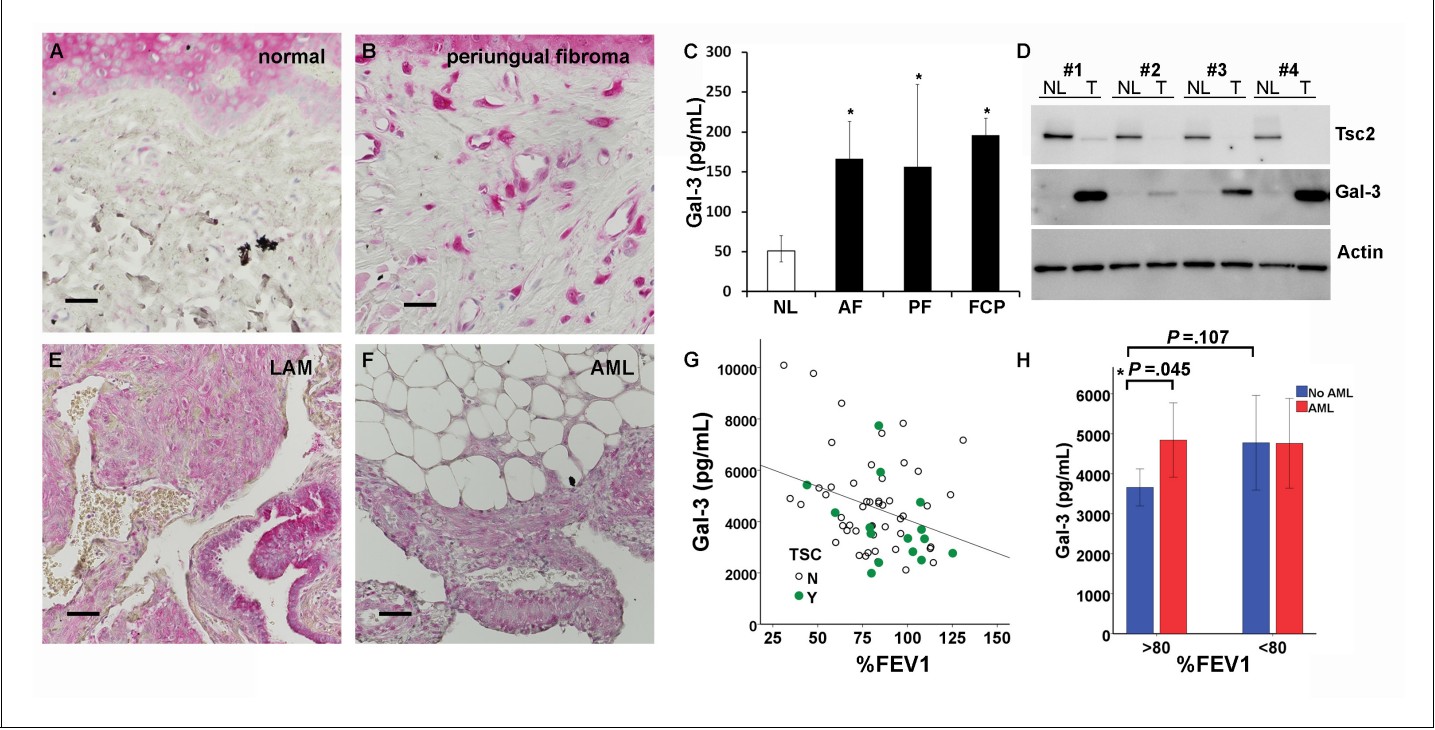

**Figure 6.** Gal-3 expression in TSC skin tumors, LAM nodules and AML of the kidney. (**A**) Gal-3 immunostaining of normal-appearing skin biopsy from TSC patient. (**B**) Gal-3 expression in TSC periungual fibroma skin tumor. (**C**) Gal-3 levels from culture supernatants of fibroblasts grown from TSC skin tumors. Normal-appearing skin from ear (NL) n = 11 patients; angiofibroma (AF) n = 7 patients, p=0.020 vs. NL; periungual fibroma (PF) n = 5 patients, p=0.028 vs. NL; fibrous cephalic plaque (FCP) n = 3 patients, p=0.026 vs. NL. For some patients, Gal-3 levels from multiple skin tumor cell lines were averaged. (**D**) Western blot showing correlation of TSC2 and Gal-3 levels. Paired skin samples from four representative patients are shown. NL = cultured normal skin fibroblasts and T = TSC skin tumor fibroblasts. (**E**) Gal-3 expression in LAM nodule representative of n = 4 LAM patient samples tested. Airway epithelium in lower left of panel is also positive. (**F**) Gal-3 expression in angiomyolipoma (AML) lesion representative of n = 3 AML patient samples tested. (**G**) Significant negative correlation of %FEV1 with Gal-3 serum levels in patients with proven LAM and not taking mTOR inhibitor (r = −0.32, p=0.010, n = 64 with only one sample per individual used for analysis). N = no TSC (sporadic LAM). Y = TSC-LAM. (**H**) Comparison of Gal-3 levels in LAM patients with and without AML. For patients with mild LAM (%FEV1 > 80), Gal-3 levels were higher in those with AML (n = 33) than without (n = 25). *p=0.045. There was no significant difference for Gal-3 levels with respect to AML status for patients with %FEV1 < 80. For all panels, * represents p<0.05. Error bars indicate ± S.D. Scale bars in A, B, E, and F are 0.05 mm. The source data for this figure are in *Figure 6—source data 1*, *2* and *3*.

The following source data and figure supplements are available for figure 6:

**Source data 1.** Source data-*Figure 6C*.
**Source data 2.** Source data-*Figure 6G and H*.
**Source data 3.** %FEV1 and Gal-3 levels in serum from LAM patients taking mTOR inhibitor.
**Figure supplement 1.** Galectin-3 expression in LAM nodule tumor cells, as well as lung epithelium.
**Figure supplement 2.** Galectin-3 expression AML tumor cells, as well as normal kidney.
**Figure supplement 3.** No correlation of %FEV1 with Gal-3 serum levels in patients with proven LAM taking mTOR inhibitor.

*Kwiatkowski et al., 2002*; *Sun et al., 2015*; *Leech et al., 2015*), but these models lacked the large thickened tortuous arteries with smooth muscle dysplasia and variable fibrosis observed in the larger vessels in Tsc2cKO[Prrx1-cre] mice. The microscopic appearance of the shoulder tumors shared features of the vascular abnormalities observed in AMLs in patients with TSC, including large dysplastic

vessels with smooth muscle hyperplasia staining positive for HMB45 and thin, fragmented elastic fibers. Additional abnormalities in these mice were similar to those observed in other models, such as liver hemangiomas, renal cystadenomas (*Kobayashi et al., 1999*; *Onda et al., 1999*; *Kobayashi et al., 2001*; *Kwiatkowski et al., 2002*) and sclerotic bone (*Fang et al., 2015a*, *Fang et al., 2015b*) in mouse models, and spleen hemangiosarcomas in the Eker rat (*Yeung et al., 1995*; *Kubo et al., 1995*). The breadth, predictability and rapid formation of multiple manifestations of tissue dysplasia with high penetrance in the Tsc2cKO[Prrx1-cre] mice make this an attractive preclinical model for TSC rather than using different mouse models for each phenotype. These mice also provide a novel model system to investigate vascular pathologies of major significance in TSC, since AMLs are a source of life-threatening hemorrhage in TSC (*Byard et al., 2003*), and aneurysms and other TSC-related vascular abnormalities can cause morbidity and mortality (*Salerno et al., 2010*).

EYFP reporter expression in the larger abnormal blood vessels in Tsc2cKO[Prrx1-cre] demonstrated that *Tsc2* deletion occurs in cells of the vessel wall and perivascular cells as expected due to embryonic expression patterns of Prrx1-cre (*Logan et al., 2002*; *Durland et al., 2008*). In addition, EYFP expression was noted in many endothelial cells, an unexpected finding based on the lack of EYFP expression observed within endothelial cells of WT Prrx1-cre expressing tissues (*Figures 2*, *Figure 2—figure supplement 6*, and *Figure 3*) and as reported elsewhere (*Greenbaum et al., 2013*). One possible explanation is that EYFP-positive mural cells with loss of Tsc2 are progenitors for abnormal-appearing endothelial cells in these enlarged vessels, consistent with the presence of populations of vascular wall progenitor cells with potential for forming endothelial cells (*Psaltis and Simari, 2015*). Two reports of *Tsc1* disruption in vascular smooth muscle did not, however, result in development of *Tsc1* KO vascular endothelial cells (*Malhowski et al., 2011*; *Houssaini et al., 2016*). It is also possible that EYFP-positive endothelial cells result from expansion of a rare population of Tsc2-deficient endothelial cells. This explanation also fits with the EYFP-positive lymphatic endothelial cells comprising the abnormal lymphatics near lymph nodes. In any case, the dramatic vascular changes and their normalization by sirolimus highlights the importance of controlled mTORC1 signaling in the development and postnatal organ homeostasis of Prxx1-expressing mesenchymal-derived tissues.

Fibroblasts grown from Tsc2cKO[Prrx1-cre] mice were used to identify a gene expression signature of *Tsc2* gene inactivation, which included genes involved in glucose metabolism, cell cycle regulation, and HIF1α responses. These processes are consistent with known regulation of these processes by loss of TSC1 or TSC2 (*Laplante and Sabatini, 2013*). Enrichment for this expression signature was tested in cancer, focusing on bladder cancer since these show mutations in *TSC1* and/or *TSC2* in about 15% of cases (*Sjödahl et al., 2011*; *Pymar et al., 2008*). Using the TCGA database of cancers, the TSC2 loss-of-function expression signature, as well as levels of *LGALS3*, were associated with bladder cancers harboring *TSC1/TSC2* inactivating mutations. We propose that TSC2 loss imparts a common transcriptional expression signature including *LGALS3* that could be considered for diagnosis and/or treatment options. Mutations in genes concurrently with *TSC1/TSC2*, such as those frequently occurring in bladder cancers (*Guo et al., 2013*), will likely define the best treatment course. Refinement of the signature based on effects of additional mutations, differing cell types and/or sirolimus sensitivity may improve the utility of this approach.

We found that Gal-3 is elevated inTSC2 deficiency and mTORC1 activation, as Gal-3 levels were increased in human TSC skin tumors and TSC2-null skin tumor fibroblasts. Gal-3 serum levels negatively correlated with severity of LAM disease and positively correlated with the presence of AML in patients with mild LAM (%FEV1 >80). The usefulness of Gal-3 as a serum marker for LAM is unclear, as serum levels in LAM patients were not different from normal subjects. It is possible that the positive correlations of serum Gal-3 levels with disease severity in LAM patients is influenced by baseline differences in Gal-3 production. Polymorphisms in *LGALS3* are known to impact Gal-3 serum levels (*Okada et al., 2006*; *Hu et al., 2011*), so future studies could test for *LGALS3* polymorphisms as a marker for rates of disease progression. Gal-3 is a pleiotropic carbohydrate-binding protein that can be located intracellular or secreted, whose expression is HIF1α-inducible (*Greijer et al., 2005*), is a known angiogenic factor (*Markowska et al., 2010*; *Nangia-Makker et al., 2000*), and frequently has altered expression in cancer (*Thijssen et al., 2015*). Gal-3 overexpression is not restricted to bladder cancer or TSC since it is highly expressed in other cancers and various fibrotic tissues (*Li et al., 2014*; *Liu and Rabinovich, 2005*) and Gal-3 serum levels provides prognostic information for heart failure (*Yancy et al., 2013*). It is not yet known whether additional TSC tumors express Gal-3 or

whether Gal-3 levels are elevated in children with TSC. Our findings that sirolimus only partially reduced Gal-3 levels in Tsc2cKO[prrx1-cre] mice is consistent with the idea that Gal-3 may reflect residual amounts of Tsc2-deficient cells during treatment; although tumors were nearly eliminated by sirolimus in the mice, Tsc2-deficient mesenchymal cells were still present and maintained higher than normal serum Gal-3 levels.

A consequence of increased Gal-3 in TSC and LAM may be the stimulation of tumor-promoting pathways. Indeed, Gal-3 is involved in stimulating angiogenesis, neoplastic transformation, resistance to apoptosis, and in metastasis (*Liu and Rabinovich, 2005*). Gal-3 levels in bladder cancer are associated with tumor proliferation, progression, and clinical outcome (*Zeinali et al., 2015*). Determining the extent to which Gal-3 is related to the pathology or progression of TSC or LAM will be instructive regarding its potential as a new therapeutic target.

## Materials and methods

### Patients

Serum and skin samples used for this study were from a cohort of 139 patients diagnosed with LAM based on a combination of clinical, histopathological, radiological, and serum VEGF-D criteria were used for this study. Patients were enrolled in protocols at the National Institutes of Health (NIH) Clinical Center (protocol 95 H-0186; 96 H-0100; 00H0051), which were approved by the National Heart, Lung, and Blood Institute Institutional Review Board and, written informed consent was obtained for each individual.

### Animal studies

Mice were housed at the USU animal facility and at the National Heart, Lung, and Blood Institute (NHLBI). All animal studies were performed in adherence to protocols that were approved by the Uniformed Services University (USU) Institutional Animal Care and Use Committee and NHLBI Animal Care and Use Committee protocol (under protocol H-0128.) Mice carrying the *Tsc2-floxed* allele (*Hernandez et al., 2007*), were a gift from Dr. Michael Gambello. Tsc2cKO[Prrx1-cre] mice were generated by crosses consisting of *Prrx1-cre+/-* males and homozygous (*Tsc2fl/fl*) females. Male and female Tsc2cKO[Prrx1-cre] mice were subfertile and therefore were not used for breeding. *Tsc2 floxed* mice were crossed with (*Rosa26) Loxp-stop-Loxp-EYFP* cre reporter mice (*Srinivas et al., 2001*) to track any cell that expressed or was derived from a Prrx1-cre-expressing cell. Mouse lines expressing the *Prrx1-Cre* (*Logan et al., 2002*) transgene and the EYFP cre reporter (*GT(Rosa)26Sor)* transgene were purchased from The Jackson Laboratory.

### Genotyping and PCR

PCR was performed on DNA isolated from earpunch samples. Mice were genotyped for *Tsc2* alleles using three primers in one PCR reaction: Fwd *Tsc2* (common): 5'-GCAGCAGGTCTGCAGTGAAT, Rev *Tsc2* (*Tsc2fl*, *Tsc2+*): 5'-GCAGCAGGTCTGCAGTGAAT, Rev (Tsc2⁻): 5'-CCTCCTGCATGGAGTTGAGT. Band sizes were *Tsc2+* (390 bp), *Tsc2fl* (434 bp) and *Tsc2⁻* (547 bp). For *Prrx1-cre* genotyping: Fwd *Prrx1-cre:* 5'-CTCCCTCCTCCTCTCTTGCT, Rev *Prrx1-cre*: 5'-CCATGAGTGAACGAACCTGGTCG. A band size of 761 bp was present for the transgene. For genotyping the EYFP *Gt (ROSA26)* reporter: Fwd *Gt(ROSA26)Sor* 5'-AAGACCGCGAAGAGTTTGTC, Rev *Gt(ROSA26)Sor:* 5'-AAAGTCGCTCTGAGTTGTTAT. PCR product sizes were 320 bp for mutant *ROSA26* locus and 600 bp for WT *ROSA26* locus. To detect *Tsc2* gene copy number, TaqMan real-time DNA copy number assays were used for *Tsc2* intron 3-exon 3 and intron 5-exon six and *Tsc1* exon 6-intron 6 (Thermo-Fisher Scientific).

### Primary fibroblast cell line isolation and culture

Neonatal mouse dermal fibroblasts: Isolation was carried out essentially as described (*Lichti et al., 2008*), except that skin from limbs was used instead of trunk skin. Each neonate was genotyped using PCR prior to cell isolation and genotyping. Cells were cultured in DMEM with 10% FBS and antibiotics.

Human skin tumor fibroblasts: Biopsies used for cell culture were cut into pieces and placed into 35 mm culture dishes with enough DMEM with 10% FBS containing antibiotics to just cover the

pieces. Adherent fibroblasts that migrated out were expanded and cryopreserved. Cell lines were all tested for tuberin (TSC2) levels and pS6 levels under serum-starved conditions by Western blot. Cells displaying decreased tuberin and TSC2 activation were used for analysis of Gal-3 levels. Mouse and human cells were free from detectable mycoplasma, using the ATCC Universal Mycoplasma Detection Kit #30–1012K.

## Gal-3 ELISA measurements

Mouse Gal-3 ELISA assays were purchased from R&D Systems (DY1197) and human Gal-3 ELISA was from eBioscience/Affymetrix (BMS279/4). Mouse cell culture supernatants were diluted 1:50, while mouse serum was diluted 1:400. For ELISA of human samples, culture supernatants were undiluted, and human serum diluted 1:10. Gal-3 levels were calculated based on a standard curve of purified recombinant Gal-3 using ELISA analysis software (http://www.elisaanalysis.com/).

## Culture supernatants

250,000 cells were seeded per well of 6-well tissue culture dishes. After 24 hr, the media were removed, cells washed 1X with PBS and media (DMEM plus 1% FBS) was added either containing 20 nM sirolimus or DMSO as a vehicle control. Media was changed after 24 hr and supernatants and cell lysates were collected after 48 hr incubation. Gal-3 levels were calculated from a standard curve and normalized to total cellular protein content.

## Sirolimus treatment

Mice were injected I.P. with sirolimus or vehicle 5 mg/kg every other day. A stock solution of sirolimus (LC Laboratories) was dissolved in 100% ethanol to a concentration of 50 mg/mL. For injection, sirolimus was suspended to a concentration of 0.5 mg/mL in a vehicle consisting of 5% Tween 80 (Sigma) and 5% PEG 400 (Sigma). Footpad thickness was measured weekly using calipers.

## Mouse MRI and in vivo fluorescence

MRI was performed in a 7T, 16 cm horizontal Bruker MRI system (Bruker, Billerica, MA) with Bruker ParaVision 5.1 software. Mice were anesthetized with 2–3% isoflurane with ECG and respiratory detection (SA Instruments, Stony Brook, NY). Mice were imaged in a 35 mm, m2m Imaging birdcage volume coil (m2m Imaging, Cleveland, OH). Magnevist (gadopentetate dimeglumine, Bayer Health-Care, Montville, NJ) diluted 1:10 with sterile 0.9% saline, was administered IV at 0.1 to 0.3 mmol Gd /kg. ECG-gated 2D spin echo images of the chest and abdomen (TR = 1000 ms, TE = 12 ms, 15–20, 1 mm slices, 100–120 micron in-plane resolution) and respiratory-gated 3D FISP images (TR = 7.72, TE = 3.35, flip angle (FA) = 15, approximately 100 x 100 × 450 micron resolution varying slightly with body size) of the whole body were acquired. 2D MR angiography of the head, abdomen and hips were acquired for selected mice (TR = 20 ms, TE = 4.2 ms, FA 90, 86–96 slices, 0.3 mm slice thickness, 82–94 micron in plane resolution). Images were analyzed with ImageJ software.

In vivo EYFP fluorescence in neonatal mice was detected using the Bruker In Vivo Xtreme imaging system (Billerica, MA). Neonates were euthanized by carbon dioxide immediately prior to imaging.

## Extraction and analysis of blood, serum and extrapleural fluid

Blood was extracted from sacrificed mice by cardiac puncture and divided between tubes for serum (BD Microtainer #365967) and blood (Sarstedt 1.3 ml K3E). Fluid was extracted from sacrificed Tsc2cKO[Prrx1-cre] mice which showed visible swelling in the shoulder/axilla region using a 3 mL syringe with 20G needle. Fluid extracted was variably pink or reddish and cloudy. Both blood and extrapleural fluid were centrifuged in serum separator tube to remove red blood cell component. Serum chemistry and CBC analysis were performed at the NIH Diagnostic and Research Services Branch, Division of Veterinary Resources.

## Immunohistochemistry and histological analysis

Sections were deparaffinized in xylene, and rehydrated through graded alcohol series using distilled water. Sections were heated for antigen retrieval in boiling 0.01 M citrate buffer pH 6.0 for 10 min or treated with 0.1% pepsin (for anti-HMB-45 only). After being washed in PBS, the tissue sections were incubated with 5% goat serum in PBS for 1 hr at room temp to block nonspecific-binding sites.

Primary antibodies (see Table S5 for details) were diluted in blocking buffer applied to tissue sections overnight at 4°C in a moisture chamber. The following day, tissue sections were washed with PBS and incubated with biotinylated secondary antibody for 30 min at room temperature, then for 30 min in avidin-biotinylated complex (Vectastain ABC kits, Vector Laboratories, Inc.) after washing. Staining was visualized with Alkaline Phosphatase substrate (Vector Laboratories) for about 30 min. Antibodies used for immunohistochemistry were: anti-GFP (Life Technologies, #A11122) 1:1000, anti-galectin-3 (Abcam, #ab53082) 1:200, anti-alpha SMA (Abcam, #ab5694)1:200, anti-CD31 (Abcam,, #ab28364) 1:30, anti-VEGFR-3 (BD Biosciences, #552857) 1:30, anti-melanosome, clone HMB-45 (Dako/Agilent Technogies).

The sections were washed thoroughly in tap water. Meyer's haematoxylin served as a counterstain. Finally, the sections were mounted in permanent mounting medium (Vector Laboratories). For Gal-3 IHC of TSC skin tumors 4 normal, 7 angiofibromas, 4 periungual fibromas, and 1 fibrous cephalic plaques were stained and analyzed. Most images were taken on a Nikon Eclipse Ti microscope with Nikon DS-Ri2 color CMOS camera. For analysis of morphology and measurements of skin thickness, H&E slides were converted into high resolution digital image files with a NanoZoomer Digital Pathology System (Hamamatsu) available in the USU Bioinstrumentation Center (BIC). NDP. view2 software was used to open NanoZoomer files and perform digital measurements of skin thickness.

## Mouse RNA sequencing and analysis

RNA was extracted from $Tsc2^{-/-}$ (KO) (n = 3), KO treated with sirolimus (n = 3), $Tsc2^{fl/fl}$ (WT) (n = 3), and WT treated with sirolimus (n=3) neonatal mouse dermal fibroblasts using an RNeasy Mini Kit (Qiagen) and on-column DNA digestion. Sequencing libraries were generated using the TruSeq Stranded mRNA Library Preparation Kit (Illumina) before assessing library size distribution using the Fragment Analyzer (Advanced Analytical Technologies) and quantity using the KAPA Library Quantification Kit for NGS (Kapa Biosystems). Sequencing was conducted using a NextSeq 500 (Illumina) with paired-end reads at 75 bp length. Sequencing data were aligned to a mouse transcript models using STAR (*Dobin et al., 2013*) and expression was quantified using RSEM (*Li and Dewey, 2011*) to obtain FPKM expression values. FPKM values were adjusted by adding 1 and applying log2 transformation. Differential expression between sample groups were calculated by two class SAM for wild type versus KO, and by paired two-class SAM for sirolimus treated versus vehicle (*Tusher et al., 2001*). Differentially expressed genes were selected as those with FDR < 10%. Genes differentially expressed between KO and WT were referred to as the TSC2 expression signature. Differentially expressed candidate transcripts were queried for cellular component and biological process enrichment analysis using PANTHER Classification System (*Mi et al., 2017*, *2013*) and ConsensusPathDB (*Kamburov et al., 2009*). Data are available through the Gene Expression Omnibus (GEO) under accession number GSE92589.

## Human gene expression analysis

The results published here are based on data generated by the TCGA Research Network: http://cancergenome.nih.gov/. Somatic mutation and gene expression quantification data of The Cancer Genome Atlas (*Cancer Genome Atlas Research Network, 2014*) were downloaded from http://firebrowse.org/ (n = 391 tumors). Expression data (normalized RSEM values) were adjusted by adding 1, applying log2 transformation, and standardized to z-scores. Expression data were reduced to those human genes contained in the mouse Tsc2 expression signature. Expression of genes in the signature that were underexpressed in KO vs WT were multiplied by −1 so that all genes in the signature are in the same direction. For each human bladder tumor, a TSC2 expression score was defined as the mean of the resulting expression values, similar a published method for calculating expression scores in human tumors from a model system (*Wilkerson et al., 2012*). Human tumors with a non-silent *TSC1* or *TSC2* mutation were considered *TSC1/2* mutant and others as wild type.

## Western blotting analysis

Cultured fibroblasts were lysed in 20 mM Tris, pH7.5, 150 mM NaCl, 20 mM NaF, 2.5 mM $Na_4P_2O_7$ × $10H_2O$, 1 mM $\beta$-glycerophosphate, 1% NP-40, 1 mM benzamidine, 10 mM 4-nitrophenyl phosphate, 0.1 mM PMSF (reagents from Sigma). Ten percent SDS-PAGE was run using 5 ug of total

protein lysate. Proteins were transferred to Invitrolon PVDF (Life Technologies). For blocking membranes, TBS/0.1% Tween 20/5% NF milk for 1 hr was used. Antibodies were diluted in blocking buffer and incubated overnight at 4°C. Antibodies which reacted with tuberin/Tsc2, p-S6 ribosomal protein (ser235/236), and total S6 ribosomal protein were purchased from Cell Signaling (#4308, #2211 and 2217, respectively). Monoclonal anti-$\beta$-actin was purchased from Sigma (#A5441).

### Isolation and culture of splenic and forepaw tumor cells

Paw tumors or approximately 0.3 g splenic tumor were excised, minced and digested for 2 hr with 0.35% collagenase type I (Worthington Biochemical) dissolved in DMEM containing 10% FBS. Cells were washed three times with PBS and cultured on gelatin-coated dishes in endothelial cell growth media (Vasculife EnGs, LifeLife Technologies) containing antibiotics and antifungals.

### Flow cytometry

CD31 and CD90.2 extracellular expression in cells derived from Tsc2cKO tumors were analyzed by flow cytometry. Cells cultured on plastic were harvested with Accutase (Innovative Cell Technologies). Cells were washed with autoMACs Rinsing Solution containing 1% BSA. After washing, $2.5 \times 10^5$ cells were labeled with APC-labeled anti-CD31 antibodies (Miltenyi Biotec Cat# 130-097-420), or APC-anti-CD90.2 (Miltenyi Biotec Cat# 130-091-790). Cells were then analyzed on a BD LSRII flow cytometer. The enrichment of YFP positive splenic cells was done by fluorescent sorting on a BD FACSAria cell sorter.

### Statistical analysis

Means are presented as mean ± standard deviation unless otherwise indicated. Parametric or non-parametric statistical analysis was chosen based on visual assessment of the normality of distribution of the data. For comparison of means between two normally distributed groups, student's t-test was used. For effect of sirolimus on footpad thickness, data were analyzed using a mixed model for repeated measures with group as a between-subjects factor, time as a within-subjects factor, and a first-order autoregressive structure for the within-subject correlation. Because the overall model was significant ($p<0.001$) and included a significant group x time interaction ($p<0.001$), follow-up ANOVA models were performed to compare sirolimus vs. vehicle at each time point with a Bonferroni adjustment for multiple comparisons. For pairwise comparison of means from non-normally distributed groups (Gal-3 secretion from patient-derived cell lines: tumor vs. normal) ELISA values were log transformed and pairwise student's t-test was used. Data were then back-transformed and plotted. For survival analysis, Kaplan Meier plots and log-rank tests were used. For correlation studies, Pearson correlation was used (2-tailed). The required sample size to detect a correlation of 0.3 or greater with 80% power and 5% significance, (two sided) is 85. The actual sample size for this study was limited to the number of patients samples that could accrued over a 6 month time period. Samples from patients without mTOR inhibitor were analyzed using either two-way factorial ANOVA or with main effects of AML (yes/no) and %FEV1 (<80 or >80) and their interaction. The simple main effect of AML was then estimated at each level of %FEV1. A multivariate linear regression model was used to identify variables that were independently associated with Gal-3. Independent variables included in this model were age, $D_{LCO}$, lymphatic involvement, AML and %FEV1. In the multivariate regression model, none of the variables tested (age, DLCO, lymphatic involvement, AML and FEV1) showed a significant association with Gal-3 after adjusting for the other variables (all P values >0.15).

## Acknowledgements

We thank Dr. Michael Gambello for providing us with *Tsc2* floxed mice. Statistical analysis of results was performed by Cara Olson, MS, DrPH, from the USUHS Biostatistics Consulting Center. Assistance with flow cytometry experiments and analysis was provided by Kateryna Lund in the USU Biomedical Instrumentation Center. Quality assurance, RNA library preparation, and RNA sequencing runs were performed by Gauthaman Sukumar (The American Genome Center, USUHS). Assistance in providing FFPE lung and kidney tissues and slides from LAM patients was provided by Gustavo Pacheco-Rodriguez. Research reported in this publication was supported by the National Institute of Arthritis and Musculoskeletal and Skin Diseases of the National Institutes of Health under award number R01AR062080 to Dr. Darling and a Collaborative Health Initiative Research Program (CHIRP)

award to Dr. Darling. JK, SAA, WKS, NN, and JM were supported by the Intramural Research Program of the National Institutes of Health/NHLBI. Content of manuscript is solely the responsibility of the authors and does not necessarily represent the official views of the National Institutes of Health, Uniformed Services University, or the Department of Defense.

## Additional information

### Funding

| Funder | Grant reference number | Author |
|---|---|---|
| National Institute of Arthritis and Musculoskeletal and Skin Diseases | NIH R01,AR062080 | Thomas N Darling |

The funders had no role in study design, data collection and interpretation, or the decision to submit the work for publication.

### Author contributions

PJK, Conceptualization, Data curation, Formal analysis, Investigation, Methodology, Writing—original draft; RLT, Data curation, Formal analysis, Investigation; JK, Resources, Data curation, Writing—review and editing; J-aW, EM, Data curation, Methodology; SAA, Data curation, Formal analysis, Writing—original draft; VH, Data curation, Formal analysis; WKS, Data curation, Project administration, Writing—review and editing; SL, Resources, Data curation, Investigation; NN, Data curation, Writing—review and editing; JDB, Conceptualization, Formal analysis, Writing—review and editing; MDW, Resources, Data curation, Formal analysis, Writing—original draft, Writing—review and editing; CLD, Resources, Data curation, Formal analysis; JM, TND, Conceptualization, Resources, Formal analysis, Supervision, Funding acquisition, Investigation, Methodology, Writing—original draft, Project administration, Writing—review and editing

### Author ORCIDs

Thomas N Darling, http://orcid.org/0000-0002-5161-1974

### Ethics

Human subjects: Patients were enrolled in protocols at the National Institutes of Health (NIH) Clinical Center (protocol 95-H-0186; 96-H-0100; 00H0051), which were approved by the National Heart, Lung, and Blood Institute Institutional Review Board and, written informed consent was obtained for each individual.

Animal experimentation: This study was performed in accordance with the standards for the care and use of animal subjects as stated in the Guide for the Care and Use of Laboratory Animals. Animal studies at the Uniformed Services University of the Health Sciences were performed using experimental (DER-14-802) and breeding (DER-14-798) protocols approved by the USUHS Institutional Animal Care and Use Committee. Animal studies at the National Institutes of Health were performed using protocol H-0128 approved by the NHLBI Animal Care and Use Committee.

## Additional files

### Major datasets

The following dataset was generated:

| Author(s) | Year | Dataset title | Dataset URL | Database, license, and accessibility information |
|---|---|---|---|---|
| Klover PJ, Thangapazham RL, Kato J, Wang J, Anderson SA, Hoffmann V, Steagall WK, Li S, | 2016 | Tsc2 KO cells have a transcriptional signature found in bladder tumors with TSC1/2 deficiency and that is corrected by sirolimus and includes Lgals3 | https://www.ncbi.nlm.nih.gov/geo/query/acc.cgi?acc=GSE92589 | Publicly available at the NCBI Gene Expression Omnibus (accession no: GSE92589) |

McCart E, Nathan N, Bernstock J, Wilkerson MD, Dalgard CL, Moss J, Darling T    overexpression

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
