## [Decision Letter]

Thank you for submitting your article "Tsc2 disruption in mesenchymal progenitors results in highly vascular tumors overexpressing *Lgals3*" for consideration by *eLife*. Your article has been reviewed by three peer reviewers, one of whom is a member of our Board or Reviewing Editors and the evaluation has been overseen by Charles Sawyers as the Senior Editor. The reviewers have opted to remain anonymous.

The reviewers have discussed the reviews with one another and the Reviewing Editor has drafted this decision to help you prepare a revised submission

Summary:

Klover et al., report a new mouse model in which Tsc2 was deleted in craniofacial and limb bud mesenchymal progenitors. This is of considerable interest and importance because mesenchymal tumors of the kidney (angiomyolipoma) and lung (LAM) are a leading cause of morbidity and mortality in TSC, and mouse models of these tumors have been very difficult to establish. Tumors develop on the forelimbs that contain adipose tissue, vascular lesions, and lymphatics, reminiscent of angiomyolipomas. While the reviewers find the mouse model of considerable interest, they feel that additional experimentation would better characterize the model and determine whether it truly resembles the tumors that develop in TSC patients (angiomyolipomas and LAM). In particular, they do not appear to have analyzed melanocytic markers, a "gold standard" of LAM and AML.

Essential revisions:

1) The histologic evaluation of these tumors could be provided in more detail. For example, how were the fluid filled structures determine to be lymphatics? Was the milky fluid within the cysts thought to be chyle? The vascular lesions should not be described as "vascular tumors" but instead as vascular anomalies within tumors. Higher power images of the fat and additional higher power examples of the vascular lesions would permit better comparison with angiomyolipomas. Did these forelimb tumors express the typical features of angiomyolipomas such as immunoreactivity with the HMB-45 antibody or expression of MiTF?

2) The number of bladder tumors with TSC1/2 mutations looks considerably higher than expected, based on the scatterplot. How were true TSC1/2 mutations in the TCGA identified, and distinguished from "passenger" variants of unknown significance?

3) No unbiased approach leading to the identification of LGALS3 as a key transcript downstream of TSC2 loss is reported. Unclear why the Authors have chosen to focus on Gal-3 out of the 1275 genes overexpressed in KO cells and underexpressed upon sirolimus treatment.

4) It is surprising that Gal-3 did not decrease with sirolimus treatment by Western, since the transcript was sirolimus-dependent. Were longer treatment times with sirolimus tried?

5) It would be helpful to see the boundary between normal kidney and AML in Figure 6, to ensure that the GAL-3 expression is higher in the AML. Expression does appear to be present in the LAM nodule but this needs to be shown alongside markers of LAM (HMB-45, for example) to be certain that this is a LAM nodule. A higher power image would also be helpful. The expression in the LAM cells is much less than in the airway epithelial cells. How many AML and LAM samples were tested?

6) Is it possible to determine whether Gal-3 is higher in the serum of LAM patients vs. healthy controls? Was it higher in patients with TSC vs. sporadic LAM? The size of Figure 6 makes it difficult to distinguish the TSC vs. no-TSC colors.

---

## [Author Response]

*Essential revisions:*

*1) The histologic evaluation of these tumors could be provided in more detail. For example, how were the fluid filled structures determine to be lymphatics? Was the milky fluid within the cysts thought to be chyle? The vascular lesions should not be described as "vascular tumors" but instead as vascular anomalies within tumors. Higher power images of the fat and additional higher power examples of the vascular lesions would permit better comparison with angiomyolipomas. Did these forelimb tumors express the typical features of angiomyolipomas such as immunoreactivity with the HMB-45 antibody or expression of MiTF?*

We appreciate the opportunity to show the remarkable histology of these tumors in more detail. The comments above focus mostly on the forelimb tumors and we agree that the forelimb tumors should be highlighted since they have some resemblance to AMLs and have not been previously reported in other mouse models. New images of the forelimb histology and IHC are provided in Figure 2, which necessitated that we move histological images of the forepaw and spleen to the supplement section (Figure 2—figure supplement 6). The new images of the forelimb show a low-power view of the shoulder region with insets to show higher power views of blood vessels, lymphatic vessels, and fat as requested (Figure 2, EI, EII, EIII, and EIV). New IHC studies were done for HMB-45 as requested. These demonstrate positive staining of vessels in the KO (Figure 2) but not wild type mouse (Figure 2), supporting its similarity to AML.

To answer the question about lymphatics, a new Figure 2—figure supplement 5 has been provided that shows abnormal lymphatic structures in the forelimbs of Tsc2cKO^Prrx1-cre^ mice. The figure includes low and high magnification H&E images of a dysplastic lymph node and connecting dysplastic lymphatics. Positive staining for lymphatic markers (VEGFR3, LYVE1) confirms that these are lymphatic endothelial cells.

New results are presented for milky fluid aspirated from axillary masses in three mice. A supplemental figure (Figure 2—figure supplement 3) shows the aspirate before and after centrifugation, and chemical analyses showed increased triglyceride and lower cholesterol levels suggesting that chylous fluid contributes to the aspirated fluid.

The text and title was changed from “vascular tumors” to “vascular anomalies”.

*2) The number of bladder tumors with TSC1/2 mutations looks considerably higher than expected, based on the scatterplot. How were true TSC1/2 mutations in the TCGA identified, and distinguished from "passenger" variants of unknown significance?*

We now indicate in subsection “Transcriptome analysis of WT and Tsc2-deficient dermal fibroblasts with or without sirolimus” that 43 out of 391 bladder tumors contained non-silent mutations in TSC1 or TSC2 which corresponds to the scatterplot in Figure 4.

A new supplemental table ([Supplementary-material SD11-data]) indicates the numbers for each type of predicted non-silent mutations in TSC1 or TSC2. Synonymous substitutions were excluded as they were considered non-pathogenic variants.

*3) No unbiased approach leading to the identification of LGALS3 as a key transcript downstream of TSC2 loss is reported. Unclear why the Authors have chosen to focus on Gal-3 out of the 1275 genes overexpressed in KO cells and underexpressed upon sirolimus treatment.*

In our search for TSC biomarkers, we combined an unbiased screen using a false discovery rate <10% with hypothesis-based approaches. The genes overexpressed by Tsc2-null cells and decreased by sirolimus were screened using PANTHER for: 1) proteins in the extracellular space (which includes secreted proteins) and 2) genes regulating signal transduction, as potential mediators of TSC pathogenesis. This narrowed our list of 1275 KO-overexpressed and sirolimus-sensitive genes to 11 candidates. This is described in the Results, subsection “Transcriptome analysis of WT and Tsc2-deficient dermal fibroblasts with or without sirolimus”. An updated Figure 4 shows the heatmap of these 11 genes in a new panel C.

*4) It is surprising that Gal-3 did not decrease with sirolimus treatment by Western, since the transcript was sirolimus-dependent. Were longer treatment times with sirolimus tried?*

We thank the reviewers for suggesting this experiment. We repeated the experiment using 48 hr sirolimus treatment for 3 KO and 3 WT fibroblast cell lines. Cells appeared healthy after this treatment and Gal-3 supernatant (ELISA) and intracellular protein levels (Western blot) from KO cells decreased to control levels, matching the *Lgals3* mRNA expression changes. The new Figure 5 now includes these results.

*5) It would be helpful to see the boundary between normal kidney and AML in Figure 6, to ensure that the GAL-3 expression is higher in the AML. Expression does appear to be present in the LAM nodule but this needs to be shown alongside markers of LAM (HMB-45, for example) to be certain that this is a LAM nodule. A higher power image would also be helpful. The expression in the LAM cells is much less than in the airway epithelial cells. How many AML and LAM samples were tested?*

A new Figure 6—figure supplement 2 shows Gal-3 staining of normal kidney adjacent to the AML. Gal-3 expression by IHC is not higher in the AML than in adjacent normal kidney, but it is higher than normally seen in blood vessels and fat. This is also relevant to the concern raised regarding Gal-3 expression in airway epithelial cells. Gal-3 is known to be highly expressed by epithelial cells, as shown also for the skin epithelium in Figure 6 and B. The increased expression is by comparison with the related cell type in normal tissue (e.g. vessel wall, dermal fibroblasts).

As requested, staining for markers of LAM including HMB-45 and SMA was performed on serial sections. These demonstrate that regions positive for LAM markers are positive for Gal-3. A new Figure 6—figure supplement 1 and Figure 6—figure supplement 2, were created to show a low-power overview of Gal-3 staining of AML and LAM, respectively, along with higher power images of H&E, Gal-3, SMA and HMB-45.

The figure legend for Figure 6 now indicates that LAM and AML samples from n = 4 and n = 3 patients, respectively, were tested.

*6) Is it possible to determine whether Gal-3 is higher in the serum of LAM patients vs. healthy controls? Was it higher in patients with TSC vs. sporadic LAM? The size of Figure 6 makes it difficult to distinguish the TSC vs. no-TSC colors.*

Gal-3 serum levels in normal volunteers (n = 26) were not significantly different from patients with LAM and not taking mTOR inhibitor. Revisions are included in Results and the Discussion now states: “The usefulness of Gal-3 as a serum marker for TSC1/TSC2 inactivation in TSC, […] so future studies could test for *LGALS3* polymorphisms as a marker for rates of disease progression.”

Gal-3 serum levels were not significantly higher in patients with TSC vs. sporadic LAM. This analysis is underpowered for TSC patients and is therefore not mentioned in this manuscript.

A potential confounding variable for serum Gal-3 levels is body mass index (BMI). BMI has been reported to positively correlate with Gal-3 levels in normal individuals. We found a significant positive correlation between BMI and galectin-3 in patients without mTOR inhibitor (r = 0.315, P = 0.013). After adjusting for BMI, the partial correlation coefficient between FEV1 and galectin-3 in patients without mTOR inhibitor was still significant (partial r = -0.366, P = 0.004), indicating that differences in BMI did not account for the observed correlation. This is now included in the Results section of the manuscript.

The size of data points in Figure 6 graph has been increased to enable easier viewing.